

# Entrainment and suspension of sand and gravel

Jan de Leeuw[1], Michael P. Lamb[1*], Gary Parker[2,3], Andrew J. Moodie[4], Dan Haught[5], Jeremy G. Venditti[5,6], Jeffrey A. Nittrouer[4]

[1] Division of Geological and Planetary Sciences, California Institute of Technology, Pasadena, CA 91125, USA
[2] Ven Te Chow Hydrosystems Laboratory, Department of Civil and Environmental Engineering, University of Illinois at Urbana-Champaign, Champaign, IL 61801, USA
[3] Department of Geology, University of Illinois at Urbana-Champaign, Champaign, IL 61801, USA
[4] Department of Earth, Environmental and Planetary Sciences, Rice University, Houston, TX 77005
[5] Department of Geography, Simon Fraser University, Burnaby, British Columbia, Canada
[6] School of Environmental Science, Simon Fraser University, Burnaby, British Columbia, Canada

*Correspondence to: Michael P. Lamb (mpl@gps.caltech.edu)

**Abstract.** Entrainment and suspension of sand and gravel is important for the evolution of rivers, deltas, coastal areas and submarine fans. The prediction of a vertical profile of suspended sediment concentration typically consists of assessing 1) the concentration near the bed using an entrainment relation and 2) the upward vertical distribution of sediment in the water column. Considerable uncertainty exists in regard to both of these steps, and especially the near-bed concentration. Most entrainment theories have been tested against limited grain-size specific data, and no relations have been evaluated for gravel suspension, which can be important in bedrock and mountain rivers, as well as powerful turbidity currents. To address these issues, we compiled a database with suspended sediment data from natural rivers and flume experiments, taking advantage of the increasing availability of high-resolution grain-size measurements. We evaluated 14 dimensionless parameters that may determine entrainment and suspension relations, and applied multivariate regression analysis. A best-fit two-parameter equation ($r^2 = 0.79$) shows that near-bed entrainment, evaluated at 10% of the flow depth, increases with the ratio of skin-friction shear velocity to settling velocity ($u_{*skin}/w_{si}$), as in previous relations, and with Froude number ($Fr$), possibly due to its role in determining bedload-layer concentrations. We used the Rouse equation to predict concentration upward from the reference level, and evaluated the coefficient $\beta_i$, which accounts for differences between turbulent diffusivities of sediment and momentum. The best-fit relation for $\beta_i$ ($r^2 = 0.40$) indicates greater relative sediment diffusivities for rivers with greater flow resistance, possibly due to bed-form induced turbulence, and smaller $u_{*skin}/w_{si}$; the latter effect makes the dependence of Rouse number on $u_{*skin}/w_{si}$ nonlinear, and therefore different from standard Rousean theory. In addition, we used empirical relations for gravel saltation to show that our relation for near-bed concentration also provides good predictions for coarse-grained sediment. The new relations are a significant improvement compared to previous work, extend the calibrated parameter space over a wider range in sediment sizes and flow conditions, and result in 95% of concentration data predicted within a factor of nine.



## 1 Introduction

Suspension of sediment by water plays a critical role in the dynamics of rivers, river deltas, shallow marine environments and submarine fans. For example, suspended sediment dominates the load of lowland rivers and builds land in subsiding river deltas and coastal landscapes (Ma et al., 2017; Syvitski et al., 2005). Transport of sediment on the continental shelf is dominated by suspension of mud and sand due to waves and currents (Cacchione et al., 1999; Nittrouer et al., 1986). Suspended sediment provides the negative buoyancy of turbidity currents that move sand and gravel to the deep sea. Suspension of gravel is important in large floods, such as outburst floods (Burr et al., 2009; Larsen and Lamb, 2016), and in steep mountain canyons (Hartshorn et al., 2002), where it can contribute to bedrock erosion (Lamb et al., 2008a). Suspended sediment transport also is important in landscape engineering, such as river restoration (Allison and Meselhe, 2010), fish habitat (Mutsert et al., 2017), and the capacity of dams and reservoirs (Walling, 2006). The balance between entrainment and deposition from suspension determines patterns of deposition and erosion in these environments and therefore controls landform morphology and stratigraphic evolution (Garcia and Parker, 1991; Paola and Voller, 2005).

Most models for sediment suspension are based on application of Rouse theory (Rouse, 1937; Vanoni, 1946),

$$\frac{C}{C_a} = \left[ \frac{\frac{H-z}{z}}{\frac{H-a}{a}} \right]^P, \tag{1}$$

where $C$ is the volumetric sediment concentration at elevation ($z$) above the bed, $C_a$ is the reference near-bed concentration at $z = a$ and, $H$ is the flow depth. Also $P$ denotes the Rouse number,

$$P = \frac{w_s}{\beta \kappa u_*}, \tag{2}$$

in which $w_s$ is the particle settling velocity, $\kappa$ is the von Karman constant of 0.41, $u_*$ is the bed shear velocity and $\beta$ is a factor that accounts for differences between turbulent diffusivity of sediment and momentum (e.g. Graf and Cellino, 2002). The Rouse equation can be derived assuming an equilibrium suspension where the upwards flux of sediment due to turbulence ($F_z$) is balanced by a downwards gravitational settling flux ($C w_s$), and where $F_z$ is parameterized using a parabolic eddy viscosity (Fig. 1) (Rouse, 1937). Application of the Rouse equation requires specification of $\beta$ and $C_a$; significant uncertainty exists in estimates of both of those parameters.

The factor $\beta < 1$ in Equation 2 is often attributed to sediment-induced stratification (Einstein, 1955; Graf and Cellino, 2002; Wright and Parker, 2004a) or to flocculation, which mainly increases the settling velocity of fine-grained sediment (e.g. Bouchez et al., 2011; Droppo and Ongley, 1994). Although more sophisticated models exist, some of which abandon the Rouse theory entirely in favor of a more rigorous turbulence model (Mellor and Yamada, 1982), calibrating $\beta$ remains a useful and tractable approach for modeling and field application (Graf and Cellino, 2002; van Rijn, 1984; Wright and Parker, 2004b). Several formulas have been proposed for $\beta$, which have very different forms (Fig. 2; Table 1b). The relations of van Rijn (1984) and Graf and Cellino (2002) propose that $\beta$ is a function of $u_*/w_s$. Wright and Parker (2004b) propose that $\beta$ is a function of reference concentration divided by slope ($C_a/S$) and Santini et al. (2019) propose that $\beta$ is a function of $u_*/w_s$ and the ratio between flow depth and bed grain size ($h/D$). Wright and Parker (2004b) suggest that the dependence on $C_a/S$ is due



to sediment-induced stratification, while others do not provide a physical rationale for the parameters in their relations. Graf and Cellino (2002) have different formulas for cases with and without bedforms, and suggest that the turbulence generated by bedform roughness results in better sediment mixing and thus in a higher $\beta$. Sediment-induced stratification and aggregation
reduce vertical mixing, which would imply $\beta < 1$. Nonetheless, some formulas (van Rijn, 1984; Santini et al., 2019) and datasets (Graf and Cellino, 2002; Lupker et al., 2011) indicate $\beta > 1$, which implies enhanced mixing of sediment relative to momentum. Field and flume data (Chien, 1954; Coleman, 1970; van Rijn, 1984) indicate that $\beta$ is often greater than unity for the coarse grain-size fraction of the suspended material (Greimann and Holly Jr, 2001). Nielsen and Teakle (2004) have shown with mixing length theory that the Fickian diffusion model that is used in the derivation of the Rouse equation is not valid for
steep concentration gradients. Instead, for coarse sediment with a high settling velocity, the steep concentration gradient makes vertical mixing more efficient, resulting in $\beta > 1$.

Compared to $\beta$, even larger uncertainty exists in predicting $C_a$. For equilibrium suspensions with steady state concentration profiles, the near-bed concentration, $C_a$, is typically thought to be a function of flow parameters, grain size and the availability of sediment on the bed (Garcia and Parker, 1991; McLean, 1992; Wright and Parker, 2004b). By definition, the equilibrium
near-bed concentration is equal to a dimensionless entrainment parameter, $E_s = F_{za}/w_s$ (Garcia and Parker, 1991). The entrainment parameter, $E_s$, can then be used to predict the upward flux of sediment in non-equilibrium conditions (Garcia and Parker, 1991). Thus, the imbalance between the upward and downward flux of sediment determines the deposition or erosion rate, $dz_b/dt$, i.e.,

$$(1 - \lambda)\frac{dz_b}{dt} = F_{za} - C_a w_s = w_s(E_s - C_a) \,, \tag{3}$$

where $\lambda$ is the bed porosity (Parker, 1978). At steady state, Equation (3) reduces to $E_s = C_a$, and thus the near-bed concentration for equilibrium suspensions can be used to find $E_s$. As such, $C_a$ is necessary both to predict the vertical distribution of suspended sediment at steady state in Eq. (1), and the rate of erosion and deposition and the transient pickup of sediment for disequilibrium suspensions in Eq. (3).

Several existing formulas for $C_a$ are applicable only to uniform bed sediment (Akiyama, 1986; Celik and Rodi, 1984; Einstein,
1950; Engelund and Fredsøe, 1976; van Rijn, 1984; Smith and Mclean, 1977). However, given the strong control of grain size on near-bed concentrations, accurate formulas likely need to make grain-size specific predictions for sediment mixtures (Garcia and Parker, 1991; McLean, 1992; Wright and Parker, 2004b). Garcia and Parker (1991) reviewed the entrainment relations that were available at that time and found that their relation and the relations of van Rijn (1984) and Smith and McLean (1977) performed best in tests against field and experimental data for sand. Since then, new entrainment relations have been introduced
by McLean (1992) and Wright and Parker (2004b) (Table 1a). In order to compare existing models of $C_a$, we varied bed stress or grain size for each relation, while holding the other parameters constant (Fig. 3a,b). Since the equations predict concentration at different near-bed reference levels, we extrapolated, using the Rouse equation with $\beta = 1$, the predicted concentrations to a common reference level at 5% of the flow depth (Fig. 3c,d). This comparison highlights the fact that the entrainment relations differ considerably in terms of dependence on median bed grain size (Fig. 3a,c) and bed stress (Fig. 3b,d). In addition, some





predictions by existing entrainment relations (Fig. 3) are unrealistic: for example, van Rijn (1984) predicts concentrations
       greater than 100% at high shear stresses, and Wright and Parker (2004b) predicts that entrainment rates increase for larger
       grain sizes (and constant bed stress) in the gravel range, which is unlikely due to the greater weight and settling velocity of
       larger particles.  None of the existing relations have been evaluated in the gravel regime.

       The existing relations also vary in their choice of the reference height, $z = a$.  For example, Einstein (1950), Engelund and
Fredsøe (1976), van Rijn (1984) and Smith and McLean (1977) used a relation for $a$ that depends primarily on grain size ($D$),
       and secondarily on $u_*/w_s$ or Shields number, $\tau_* = \tau_b/(\rho_s - \rho_w)gD$ (where $\tau_b$ is bed stress, $\rho_s$ is sediment density and $\rho_w$ is
       fluid density) on the basis that these formulas capture the height of the near-bed saltation layer.  In contrast, Garcia and Parker
       (1991) and Wright and Parker (2004b) used a height that is a small fraction of the flow depth, and they proposed $a = 0.05H$ as
       a useful reference height, with no dependence on $D$.

We revisited the problem of sediment entrainment and suspension of cohesionless bed sediment by compiling a large database
       of sediment-size specific data for bed-sediment mixtures, testing existing relations against the database, and proposing
       improved relations for $E_s$ and $P$. Our database is a significant improvement compared to data used in past studies due to
       development of high resolution grain-size measurements using laser diffraction, which is now commonly used in field and
       laboratory studies (Lupker et al., 2011; Gitto et al., 2017; Santini et al., 2019).  These grain-size measurements allow a single
concentration profile to be separated into many grain-size specific concentration profiles and for the parameterization to be
       tested over a wide range of parameter space. Our dataset also contains a wide range of median grain sizes, extending into the
       silt regime (median bed sizes range from 44 μm to 517 μm), and expands on the number of field measurements compared to
       previous efforts.  We are unaware of studies on $C_a$ for gravel, but workers have measured saltation heights, velocities and
       bedload fluxes for gravel, and calibrated relations for these variables (Chatanantavet et al., 2013; Sklar and Dietrich, 2004).
We used existing relations for gravel and Rouse theory to check for consistency between sand suspension data and what might
       be expected for near-bed gravel concentrations.

## 2 Methods

### 2.1 Suspended sediment data

       Based on previous theory, we searched for available datasets from rivers and flume experiments that had suspended sediment
profiles $C(z)$, depth-averaged flow velocity ($U$), flow depth ($H$), channel bed slope ($S$) and bed material grain-size ($D$), and the
       grain-size distribution of the bed material and suspended sediment samples (Table 2; Table S1). Some of the experimental
       studies used a narrow grain-size distribution, and, like previous workers, we assumed that the sediment distribution was
       uniform. Many of the older datasets were also used in empirical regressions from previous relations (e.g. Garcia and Parker,
       1991; Haught et al., 2017; van Rijn, 1984; Wright and Parker, 2004b). In addition, we used a river dataset from the Yellow
River (Moodie, 2019), which provides a fine-grained end member.  In total, our database contains 180 concentration profiles
       from 8 rivers and 62 profiles from 6 different experimental studies.  We analyzed only the grain fractions coarser than 62.5



μm (i.e., sand). The mud fraction was present on the bed only in small amounts, and following previous work, mud was assumed to require a different approach, potentially due to supply limitation (Garcia, 2008), cohesion or flocculation.

Grain-size distributions in older studies were determined from sieve analysis of bed material and suspended sediment samples.

The more recent studies used laser diffraction techniques, which have the advantage that a larger number of grain-size classes can be distinguished. We calculated the grain-size specific suspended sediment concentration ($C_i$) using

$$C_i = f_i C_{tot}, \tag{4}$$

where $f_i$ is the fraction of grains of the $i$-th size and $C_{tot}$ is the total suspended sediment concentration for all sizes. In addition, we computed the $D_{50}$ (median grain size) and $D_{84}$ of the bed material using linear interpolation between the logarithm of $D$ and

the cumulative size distribution.

Concentration profiles in the database typically contain 3 to 8 measurements in the vertical dimension. The Rouse profile was fitted to the profile data for each grain-size class in log-transformed space using linear least squares (Fig. 4a). Confidence bounds (68%; 1 $\sigma$) for the fitted coefficients were obtained using the inverse R factor from QR decomposition of the Jacobian. Data were excluded from further steps in the analysis if the ratio between the upper and lower bound of the confidence interval

was greater than 10 or smaller than 0.01, as these data do not follow a Rouse relation for unknown reasons (e.g., measurement error), and would appear as sparse outliers. Of the data points analyzed, 201 points (15%) were excluded based on these criteria.

We used the Rouse equation (Eq. 1) written for each grain-size class, to extrapolate or interpolate the concentration to a reference level at 10% of the flow depth ($C_{ai}$). Extrapolation to very near the bed can be difficult due to large concentration

gradients and poorly constrained near-bed processes such as interactions with bedforms. However, a reference level that is too far away from the bed may poorly capture the exchange of sediment with the bed. Previous researchers used reference levels that were either at some fraction of the flow depth (Akiyama, 1986; Celik and Rodi, 1984; Garcia and Parker, 1991; Wright and Parker, 2004b) or that were related to the bed roughness height (Einstein, 1950; Engelund and Fredsøe, 1976; Garcia and Parker, 1991; Smith and Mclean, 1977). We explored the collapse of the entrainment data for both types of reference levels.

For each reference level, we fit our preferred and best-fitting two-parameter entrainment relation (as described in the Results) to the data with $u_{*skin}/w_{si}$ as its first parameter and Froude number as the second parameter. We found that a reference level at a fraction of the flow depth gave a better collapse of the data than a reference level related to the saltation layer height. Furthermore, we also tested different flow-depth fractions and found that the fit improved, in a least-squared sense, as the reference level moved to a larger fraction of the flow depth (Fig. 4B). However, there is little change in $r^2$ once the reference

level is higher than ~10% of the flow depth. Therefore, we used a reference level at 10% of the flow depth for all results shown below.

For sediment mixtures, the grain-size specific near-bed concentration is partially controlled by the fraction of each grain-size class in the surface bed material. To account for this effect, Garcia and Parker (1991) introduced an entrainment rate ($E_{si}$) for each grain-size class that is linearly weighted by the fraction of that material in the bed:



$$E_{si} = \frac{C_{ai}}{F_{bi}}, \tag{5}$$

where $C_{ai}$ is the near-bed concentration of that grain-size class and $F_{bi}$ is the fraction of bed material that falls in that grain-size class. For uniform sediment, the entrainment rate ($E_{si}$) is equivalent to the near-bed concentration ($C_{ai}$).

## 2.2 Independent parameters and fitting approach

Here we review the independent parameters that we evaluated for dependencies with entrainment, $E_{si}$, and for $\beta$ in the Rouse

number. The primary group of parameters describes the ratio between bed stress and grain size or grain settling velocity. These parameters include the ratio between shear velocity and settling velocity ($u_*/w_s$), where we evaluated the total shear velocity as,

$$u_* = \sqrt{\tau_b/\rho_w} = \sqrt{gHS}, \tag{6}$$

assuming steady, uniform unidirectional flow.  Others have proposed that entrainment depends on the skin-friction portion of

the total shear stress (minus the portion due to form drag), and we used the Manning-Strickler relation to calculate the skin-friction shear velocity as,

$$u_{*skin} = \left(\frac{U}{8.1}\right)^{0.8} (gSk_s)^{0.1}, \tag{7}$$

where $k_s = 3\,D_{84}$ is the grain roughness on the bed. To calculate the particle settling velocity, we followed Ferguson and Church (2004) for each grain-size class,

$$w_{si} = \frac{RgD_i^2}{C_1\nu+\left(0.75C_2RgD_i^3\right)^{0.5}}, \tag{8}$$

in which $R = (\rho_s - \rho_w)/\rho_w$ is the submerged specific density of sediment, $\nu$ is the kinematic viscosity of the fluid, $C_1 = 18$ and $C_2 = 1$ are constants set for natural sediment, $D_i$ is the grain diameter within the size class of interest.  Another parameter that relates to the ratio between bed stress and gravity acting on the grains is the Shields number,

$$\tau_* = \frac{\tau_b}{(\rho_s-\rho)gD_i}, \tag{9}$$

and we again assumed steady, uniform flow to find $\tau_b = \rho gHS$. Similar to the shear velocity, it is also possible to calculate a Shields number for the skin-friction component of the total shear stress,

$$\tau_{*skin} = \frac{\tau_{skin}}{(\rho_s-\rho)gD_i}, \tag{10}$$

where $\tau_{skin} = \rho u_{*skin}^2$ by definition.  Shields numbers can be rewritten in terms of $u_*/w_{si}$ through use of a particle drag coefficient,

$$C_d = \frac{RgD_i}{w_{si}^2}, \tag{11}$$

which we also evaluated.

The next group of parameters describes dimensionless particle sizes, including the particle Reynolds number,

$$R_p = \frac{u_*D_i}{\nu}. \tag{12}$$



Likewise, this parameter can also be calculated with the skin-friction component of the shear velocity,

$$R_{p,skin} = \frac{u_{*skin}D_i}{\nu}.$$
(13)

A particle Reynold number can be defined without shear velocity as,

$$Re_p = \frac{\sqrt{RgD_i}D_i}{\nu}.$$
(14)

For sediment mixtures, the relative particle size might play a role due to hiding and exposure effects (Garcia and Parker, 1991; Wright and Parker, 2004b); this effect can be captured with $\left(\frac{D_i}{D_{50}}\right)$.

Sediment-induced density stratification can decrease entrainment by dampening near-bed turbulence, and this effect is thought to be most important in deep, low gradient rivers (Wright and Parker, 2004b). Wright and Parker (2004b) proposed that the ratio of near-bed concentration to bed slope is a good predictor for stratification, $\frac{C_a}{S}$, where they used $C_a$ at 5% of the flow depth. Large, low gradient rivers also have small Froude numbers and low bed slopes, so we evaluated Froude number and slope as additional parameters. Froude number was calculated as:

$$Fr = \frac{U}{\sqrt{gH}}.$$
(15)

The entrainment rate could also be affected by turbulence or changes to the boundary layer from bed roughness or bedforms, which tend to correlate with a flow resistance friction coefficient (e.g., Engelund and Hansen, 1967),

$$C_f = \frac{u_*^2}{U^2}.$$
(16)

In order to find relations that explain the variation in the data for $E_{si}$ and $\beta$, we regressed the data against the 14 independent

variables above. In some applications, like reconstructing flow conditions from sedimentary strata, it is useful to have an entrainment relation that depends on $u_{*skin}$, while for most forward modeling a relation based on $u_*$ is preferred. The two shear velocities are highly correlated; therefore, we explored two versions of the fit relations using either $u_*$ or $u_{*skin}$, but not both at the same time. Because the Rouse parameter ($P_i$) by definition depends on $u_*/w_{si}$ (or $u_{*skin}/w_{si}$) (Eq. 2), we found best-fit relations for $P_i$ rather than $\beta_i$ to avoid spurious correlation, and then solved for $\beta_i$ using those relations and Eq. (2).

Some studies (Bennett et al., 1998; Muste et al., 2005) have shown that $\beta$ can vary considerable over the flow depth, but this effect cannot be incorporated in the Rouse solution; instead we find one value of $\beta_i$ that best fits the concentration profile for each grain size class.

We started the analysis by testing all models with one explanatory variable and rank the models according to the coefficient of determination from linear least squares regression ($r^2$) evaluated in log-log space. Next a second parameter was added and

the resulting two-parameter models were ranked according to $r^2$. The procedure was repeated with additional parameters until the increase in $r^2$ was smaller than 0.04. For the fitting of multiparameter models, we varied the exponents on each parameter in the model simultaneously to find the combination of exponents that yielded the best fit. This approach gave a higher $r^2$ compared to the stepwise approach used in previous work (e.g., Garcia and Parker, 1991) of first fitting the dominant variable and then fitting the secondary variables to the residuals. In addition, we tested fitting with the York method, which gives less





weight to data with large errors (York, 1968). All parameters were used to evaluate relations for $E_{si}$ and $\beta$, except for $\left(\frac{D_i}{D}\right)$ we

only use for $E_{si}$ since it is relevant for particle-particle interactions in the bed. In the results we report two versions of the best

fitting one, two and three parameter models: one that based on the total bed shear stress and one that is based on the skin-

friction component of the bed shear stress. Model fits using all possible combinations of the input parameters are given in

Table S2.

## 2.3 Comparison to theory for gravel

Although gravel suspension is important in bedrock and steep mountain rivers, and during large floods, we are not aware of

datasets of suspension in the gravel range. Following previous work (McLean, 1992; Lamb et al., 2008a) our approach was to

derive the near concentration in the bedload layer, and then use Rouse theory to predict that concentration at $0.1H$ to compare

with the sand datasets.

The near-bed concentration within the bedload layer can be calculated by continuity as

$$C_b = \frac{q_b}{(H_b U_b)},$$   (17)

where $q_b$ is the volumetric bedload flux per unit width, $H_b$ is the bedload layer thickness and $U_b$ is the bedload velocity. Most

relations for bedload flux take the form

$$\frac{q_b}{\sqrt{(RgD^3)}} = a(\tau_* - \tau_{*c})^b,$$   (18)

where $a$ and $b$ are empirical constants, which we set to $a = 5.7$ and $b = 1.5$ (Fernandez Luque and van Beek, 1976), and $\tau_{*c}$ is

the critical Shields number at initial motion, which we set to Lamb et al., (2008b)

$$\tau_{*c} = 0.15 S^{0.25}.$$   (19)

The bedload layer height and velocity were determined from Chatanantavet et al. (2013). They compiled a large dataset of

gravel saltation studies and found a good fit with the following relations:

$$\frac{U_b}{U} = 0.6,$$   (20)

$$\frac{H_b}{H} = 0.6 \left( Fr \left(\frac{D_i}{H}\right)^2 \right)^{0.3}.$$   (21)

Note that others have proposed that $U_b$ and $H_b$ depend on $\tau_*$ rather than $U$ (Sklar and Dietrich, 2004), but Chatanantavet et al.

(2013) found a better collapse using flow velocity as a scaling parameter.

Equations (17) – (21) were combined with a flow resistance relation (Eq. 7, assuming no form drag) and $C_d = 0.7$ for gravel

(Lamb et al., 2017) to calculate $C_b$ in the bedload layer as a function of only $\tau_*$ and $Fr$ using an iterative procedure. To predict

$C_a = C (z = 0.1H)$, we extrapolated the concentration profile (Fig. 1b) from the top of the saltation layer to $0.1H$ using the

Rouse equation (Eq. 1). To obtain the Rouse number, we used our best-fit one-parameter model that uses total shear velocity

($u_*$). We then used a wide range of input parameters ($0.1 < Fr < 1$) and ($1 < \tau_* < 1000$), relevant to suspension of gravel in

mountain rivers and large floods, to predict a range of expected values of $C_a$.





## 3 Results

### 3.1 Rouse number

Figure 5 and Table 3 show the best fitting one, two and three parameter models for Rouse number. The predictions of the best-fit two-parameter model are significantly better than the predictions of the one-parameter model ($r^2 = 0.33$ vs $r^2 = 0.40$). Using a Rouse number with a constant $\beta = 0.94$ also provides a reasonably good fit and an improvement over several more involved relations (Fig. 5). Going from a two-parameter model to a three-parameter model brings a smaller improvement ($r^2 = 0.40$ vs $r^2 = 0.43$). Therefore, we recommend the two-parameter model for combined accuracy and simplicity:

$$P_i = 0.145 \left(\frac{u_{*skin}}{w_{si}}\right)^{-0.46} C_f^{-0.3}, \tag{22}$$

The relation indicates that sediment is better mixed in the water column with larger $u_{*skin}/w_{si}$ and with larger bed roughness coefficient ($C_f$). This equation can be rewritten for $\beta_i$ by combining Eq. (2) and (22), and by assuming that $u_*$ in Eq. (2) is actually $u_{*skin}$, as

$$\beta_i = 17.24 \left(\frac{u_{*skin}}{w_{si}}\right)^{-0.54} C_f^{0.3}, \tag{23}$$

Equation (22) performs well compared to previous relations, as is shown by a boxplot of measured-to-predicted ratios (Fig. 6). For the best-fit two-parameter model, the measured-to-predicted ratio falls between 0.74 and 1.29 for 50% of the data.

Because some of the dimensional quantities appear in multiple dimensional variables, and because the dimensional variables are not necessarily independent from each other, we tested for spurious correlations by rearranging the two-parameter relation for $\beta_i$ to isolate the dimensional dependencies on grain size and skin-friction shear velocity (Fig. 7). The data and our model show a decrease in suspended sediment mixing in the water column with larger grain sizes (Fig 7a), which makes physical sense. Previous relations show the same trend, but the relations of van Rijn (1994) and Graf and Cellino (2002) have stronger dependencies that are not consistent with the data. Suspended sediments are better mixed with increasing skin-friction shear velocity (Fig. 7b), as expected. However, our relation and the data suggest that $P_i$ varies proportionally to $\sim u_*^{-0.4}$ whereas standard Rouse theory (Eq. 2) indicates that $P_i$ is proportional to $u_*^{-1}$.

### 3.2 Near-bed entrainment parameter

Of all relations tested (Table 4; Table S2), the following one-parameter relation gives the best fit with the data for near bed entrainment parameter:

$$E_{si} = 4.23 \times 10^{-5} \left(\frac{u_{*skin}}{w_{si}}\right)^{1.94}. \tag{25}$$

Figure 8a shows that Eq. (24) explains a significant part of the variation in the data ($r^2 = 0.61$). Next, we tested if an entrainment relation with two variables improved the fit. Froude number was the most significant second parameter:

$$E_{si} = 4.74 \times 10^{-4} \left(\frac{u_{*skin}}{w_{si}}\right)^{1.77} Fr^{1.18}. \tag{25}$$





This two-parameter relation has a significantly better fit ($r^2 = 0.79$) with the data than the best fitting one-parameter relation
(Fig. 8b). Addition of a third variable to the model gives little further improvement of the fit (Fig. 8c; $r^2 = 0.80$). With Eq. (25),
the majority (80%) of the entrainment data is predicted within a factor of 3 (Fig. 8b; Fig. 9).

Along with our proposed new relation (Eq. 25), we also compared the dataset against the relations presented by Garcia and
Parker (1991) and Wright and Parker (2004b). The boxplots in Figure 9 highlights that some relations systematically
underpredict (Wright and Parker, 2004b) or overpredict entrainment rate (Garcia and Parker, 1991). In addition, previous
relations have a larger spread in measured-to-predicted ratios than Eq. (25).

To check for spurious correlation in the dimensionless variables, we rearranged our two-parameter entrainment relation (Eq.
25) to isolate the dependencies of entrainment on grain size (Fig. 10a) and skin-friction shear velocity (Fig. 10b). Our relation
indicates that entrainment depends on grain size to the -3.1 power in the sand range, whereas previous relations suggest a much
weaker dependence. On the other hand, compared to previous relations, our relation suggests a relatively weak dependence on
skin-friction shear velocity ($E_{si} \propto u_{*skin}^{1.77}$).

Similar to Garcia and Parker (1991) and Wright and Parker (2004b), we modified our equation such that the predicted
entrainment rate is limited at 0.3, as total suspended sediment concentrations greater than that are not physically reasonable
for dilute, turbulent flows. In addition, a threshold, best fit by eye, was added to the entrainment relation because the
concentration data falls below the trend of the regression relation at the lower flow strengths, possibly due to a threshold of
significant sediment entrainment at the reference level. The resulting equation has the following form:

$$E_{si} = \frac{4.74\times10^{-4}\left(\left(\frac{u_{skin}^*}{w_s}\right)^{1.5} Fr - 0.015\right)^{1.18}}{1+3\left(4.74\times10^{-4}\left(\left(\frac{u_{skin}^*}{w_s}\right)^{1.5} Fr - 0.015\right)^{1.18}\right)} \quad , \tag{26}$$

### 3.3 Predicting sediment concentration

In Sections 3.1 and 3.2 we found the best-fit models for entrainment ($E_{si}$) and Rouse number ($P_i$). However, ultimately, we
want to predict sediment concentration throughout the water column, and the best-fit models for ($E_{si}$) and ($P_i$) do not necessarily
combine to yield the best-fit model for sediment concentration, owing to non-linearity in $E_{si}$, $P_i$ and the Rouse equation (Eq.
1). Here we used different combinations of our preferred one-, two- and three-parameter models for entrainment ($E_{si}$) and
Rouse number ($P_i$) to predict the grain-size specific concentrations at each data point in the water column for all our entries in
the database, and assessed model performance (Table 5).

The concentration predictions improve as more parameters are added to the entrainment model, whereas a Rouse model with
more than one parameter makes the predictions worse (Table 5). Our preferred model for depth-averaged concentration uses
a two-parameter model to predict the entrainment rate and a one-parameter model for the Rouse number (Fig 11b). Such a
model gives significantly better predictions than the most basic formulation (Fig. 11) that uses one-parameter models for
entrainment and Rouse number. The goodness of the concentration predictions is fairly constant over the height of the flow.



The predictions from the upper 1/3 of the flow have a slightly lower $r^2$ (0.80) than the predictions from the lowest 1/3 of the
flow ($r^2$ = 0.89).  Ninety five percent of the data are predicted within a factor of 9.

### 3.4 Extension to gravel

The suspended sediment data that we used to calibrate the entrainment relation covers material in the sand range. To evaluate
how our entrainment relation performs for coarser suspended sediment, we used the empirical saltation equations for gravel to
infer bedload-layer concentrations, $C_b$, and interpolated these to the reference level ($0.1H$) to infer $C_a$ for gravel (Section 2.3).
Importantly, the gravel concentrations at the reference height can be predicted from the saltation model (Section 2.3) using
only the independent parameters of $Fr$ and $\tau_*$, similar to our best fitting two-parameter entrainment model (Eq. 26). The
marked parameter space on Fig. 12 shows the expected range of $C_b$ and $C_a$ for a wide range of model input parameters: $0.1 <$
$Fr < 1$ and $1 < \tau_* < 1000$. Predicted concentrations in the gravel saltation layer are up to several orders of magnitude higher
than the predictions from our entrainment relation at 10% of the flow depth (Fig. 12). However, due to the rapid decrease of
sediment concentration away from the bed predicted by the Rouse profile, the concentration inferred at $0.1H$ for gravel overlaps
with the empirical relation for sand, implying that Eq. (25) might also be a good predictor of gravel entrainment.

### 4 Discussion

### 4.1 Physical Rationale for Model Dependencies

Our preferred model for $\beta_i$ is the result of regression against a large dataset. The explanatory variables in the model are
$u_{*skin}/w_{si}$ and $C_f$, and these parameters could reflect different effects that cause a difference between the eddy viscosity and
diffusivity of sediment (Bennett et al., 1998). Compared to standard Rouse theory where the Rouse parameter is inversely
dependent on $u_{*skin}/w_{si}$ (or $u_*/w_{si}$), our results indicate a significant non-linearity. Our one-parameter model indicates that
concentration profiles are better mixed than standard theory for small $u_*/w_{si}$ and are more stratified for larger $u_*/w_{si}$, with a
transition point at about $u_*/w_{si} = 7$, corresponding to $\beta_i = 1$. Sediment-induced stratification is often cited (Winterwerp, 2006;
Wright and Parker, 2004b, 2004a) as a factor that decreases mixing of sediment (i.e. $\beta < 1$). This effect is particularly important
when absolute concentration is high, and may help explain why our best-fit model is more stratified than standard Rousean
theory for large $u_*/w_{si}$.  An alternative hypothesis by Nielsen and Teakle (2004) is that a negative dependence of $\beta$ on $u_*/w_s$
can be the result of a mixing length effect. This effect is related to the fact for steep concentration gradients, the size of the
turbulent eddies is large relative to the mean height of sediment in the flow. Under these circumstances, Fickian diffusion,
which is assumed in the Rouse derivation, is no longer appropriate to describe turbulent mixing of sediment. Instead, the large
eddies more effectively mix sediment that is concentrated close to the bed. This effect may explain the better mixed
concentration profiles we observed (i.e., $\beta_i > 1$), as compared to standard theory, at small $u_*/w_{si}$ when near-bed concentration
gradients are large.





Our relation also implies that $\beta_i$ correlates positively with $C_f$, which could be from bedforms; bedforms increase $C_f$ due to form drag (Engelund and Hansen, 1967), and they may also increase the vertical mixing of sediment by deflecting transport paths up the stoss side of the bedform and mixing suspended sediment in the turbulent wake in the lee of the bedform. Similarly, Santini et al. (2019) found that $\beta$ correlates positively with $H/D$, which is another measure for bed roughness in flows without bedforms. In agreement, Graf and Cellino (2002) reviewed a number of experimental studies and found that $\beta < 1$ for all experiments without bedforms and $\beta > 1$ for all experiments with bedforms. Flow resistance ($C_f$) can also be smaller in flows

with sediment-induced stratification, which also correlate with smaller $\beta_i$ (e.g., Wright and Parker, 2004a)

Our new relation suggests that only $u_{*skin}/w_{si}$ (or $u_*/w_{si}$) and $Fr$ are needed to predict entrainment rate, similar to the forward model we developed for the entrainment of gravel (Section 2.3). The ratio $u_{*skin}/w_{si}$ describes the fluid forces relative to gravitational settling; similar parameters have appeared in all previous relations that we reviewed (Table 1a). The reason for the increase in entrainment with Froude number is less clear. A small Froude number implies a deep and low-gradient flow,

and Froude numbers are typically smaller in natural rivers compared to flume experiments. Wright and Parker (2004b) introduced an entrainment relation with a bed slope dependency and argued that entrainment in large low sloping rivers is reduced due to stratification effects. Sediment-induced stratification causing damping of near-bed turbulence might be the cause of the $Fr$ dependency in our relation. Regardless, we found a better fit with the data using $Fr$ than using $\frac{c_a}{S}$ or $S$ (Table S2), the parameters suggested by Wright and Parker (2004b). Froude number might also influence the size and shape of

bedforms (Vanoni, 1974), which can affect boundary layer dynamics and near-bed turbulence, and surface waves. However, as discussed in Section 4.3, $Fr$ emerges as a controlling parameter in our forward model because of its role in determining bedload layer concentrations.

Many of the dimensionless parameters we evaluated correlate with each other in rivers. While $u_{*skin}/w_{si}$ or $u_*/w_{si}$ were consistently the dominant variables, several of the possible secondary variables had nearly equivalent explanatory power as

the ones given in our preferred models. For near-bed entrainment some of the more highly ranked secondary variables include $D$, $\tau_{*skin}$ and $R_p$, and for Rouse number they include $D$, $R_{p,skin}$ and $\tau_*$ (Table S2). In addition, there is some systematic deviation between datasets for different rivers, and different parameters and exponents might better minimize residuals at specific locations.

Some previous entrainment relations (Garcia and Parker, 1991; Wright and Parker, 2004b) contain an additional parameter,

$D_i/D_{50}$, that accounts for hiding and exposure effects due to nonuniform bed material. We did not find a significant improvement of fit with such a parameter.

## 4.2 Predicting sediment concentration

Combining the predictions of the grain-size specific reference concentration and Rouse parameter allows for calculation of the grain-size specific sediment concentration throughout the water column. Our relation shows that the grain-size specific

sediment concentration at any given elevation can be predicted to relatively high accuracy ($r^2 = 0.87$) using the preferred




combination of a two-parameter entrainment parameter relation and the one-parameter Rouse number relation (Section 3.3). However, it is unclear how to evaluate the sediment concentration below the reference level, which could constitute a significant portion of the load. One approach might be to assume that sediment concentration is uniform below the reference level, as might be the case in the well-mixed bedload layer (e.g., McLean, 1992). However, this assumption is inconsistent

with our analysis of saltation equations, which shows that bedload layers should have greater concentrations and be less than 10% of the flow depth (Fig. 12), especially for sand in deep flows. An alternative approach is to use the Rouse profile to extrapolate towards the bed or towards the top of the bedload layer; however, this approach is also problematic because the Rouse profile predicts infinitely large concentrations at the bed.

Some of our datasets had concentration measurements below the reference level (Fig. 13). For 22% of the measurements below

the reference level, the concentration exceeds two times the reference concentration, but there is not a clear trend of increasing concentration below the reference level. For lack of a better approach, we suggest treating the sediment concentration as constant below the reference level, and the average of our data indicates a concentration of $1.9C_{ai}$ in that layer (Fig. 14).

### 4.3 Extension to gravel and bedload-layer theory

The transport of sand and gravel are often modeled using different empirical formulas, which hinders modeling of systems of

mixed gravel-sand transport (Wilcock and Crowe, 2003) and gravel-sand transitions (Lamb and Venditti, 2016; Paola et al., 1992). Gravel also can be in suspension in bedrock and steep mountain rivers, and during large floods (Hartshorn et al., 2002; Larsen and Lamb, 2016), and it would be useful to have an entrainment relation to model sediment transport and bedrock erosion in these settings (Lamb et al., 2008a; Scheingross et al., 2014). Our entrainment relation for sand matches expectations from gravel saltation models, suggesting that the entrainment relation may be used for sand or gravel, or mixtures of the two.

However, we currently lack data of gravel suspension profiles, and developing methods to acquire such data should be the focus of future efforts.

The good fit between the modeled gravel concentrations and the measured sand data for near-bed sediment concentration suggests that the bedload layer equations (Section 2.3) might also be used to generate a forward model for near-bed sediment concentration that works for sand and gravel systems, similarly to previous efforts (e.g., McLean, 1992). To evaluate this

possibility, we used the saltation equations (Eqs.17-21) to calculate sediment concentrations within the bedload layer for conditions corresponding to our dataset entries of sand bed rivers. To extend equation (18) to grain-size mixtures, we let $D = D_i$ and $\tau_{*c} = \tau_{*ci}$ and used a hiding function (Parker et al., 1982) so that

$$\tau_{*ci} = \tau_{*c50} \left(\frac{D_i}{D_{50}}\right)^{-\gamma}, \tag{27}$$

where $\tau_{*c50}$ is the critical Shields number for the median sized bed sediment. Equation (19) accounts for hiding of smaller

grains in between larger grains, which renders the smaller grains less mobile, and the exposure of larger grains into the flow, which renders them more mobile. For $\gamma = 1$, all grains in a bed mixture move at the same bed stress, while $\gamma = 0$, the critical





stress for motion is proportional to grain weight. Gravel bedded rivers typically have $\gamma = 0.9$ (Parker, 1990). For sand, we evaluated the critical Shields number following Brownlie (1981)

$$\tau_{*c50} = 0.5\left[0.22\, Re_p^{-0.6} + 0.06 * 10^{\left(-7.7 Re_p^{-0.6}\right)}\right], \tag{28}$$

rather than Eq. (19) which is for gravel. We then calculated the concentrations at 10% of the flow depth using the Rouse equation (Eq. 1) with the best-fit one-parameter model for the Rouse number. Although the saltation equations in Section 2.3 were calibrated for gravel, similar relations are also used for sand (e.g., Lamb et al., 2008a).

Surprisingly, the near-bed sand concentration for the entries in our database are accurately predicted by the bedload forward model without any parameter fitting (Fig. 14b). In fact, the predictions by the bedload-layer forward model are only slightly

worse than the predictions by our preferred two-parameter entrainment model (Fig. 14a) that was fit to the data ($r^2$ of 0.68 vs 0.87). The model might likely be improved by accounting for bedform induced form drag. Importantly, the forward model yields the same controlling parameters as the empirical model, namely Shields number, Froude number and the bed grain-size distribution. The forward model suggests that the dependence of $E_{si}$ on $u_{*skin}/w_{si}$ (or $u_*/w_{si}$) is predominantly because larger $u_{*skin}/w_{si}$ correlates with larger $\tau_*$ and larger bedload-layer concentrations (Eq. 17 and 18), as well as more efficient mixing

of the bedload-layer sediment up to the reference height (Eq. 1). Less intuitive is the dependence of $E_{si}$ on Froude number. In the forward model, this dependence is because larger Fr, for the same $\tau_*$ correlates with larger $D_i/H$ (which can be shown by manipulating Eq. 7). In turn, larger $D_i/H$ with constant $\tau_*$ correlates with greater bedload fluxes ($q_b \propto D_i^{3/2}$ in Eq. 18), smaller bedload layer heights (due to smaller $H$ in Eq. 21) and slower bedload layer velocities (due to smaller $U$ in Eq. 20), all of which increase sediment concentration in the bedload layer (Eq. 17). Thus, the forward model indicates that the Fr-dependency on

$E_{si}$ emerges because bedload layer dynamics depend on $U$ and $H$, and explanations for this dependency that rely on stratification or surface waves are not necessary. The bedload-layer model, while slightly more complicated to implement, may provide a more robust solution when working outside of parameter space used to derive the empirical model, since it has a more physical basis. For example, the forward model can explicitly account for gravity and other physical properties of the sediment and fluid. In addition, the bedload-layer model allows a more mechanistic link between the bedload and suspended load, and avoids

uncertainty in how to evaluate the sediment concentration below the reference level. That said, more accurate predictions are still achieved with the empirical entrainment relation.

## 5 Conclusions

We proposed new empirical models for the entrainment of bed material into suspension and for the shape of the concentration profile above a near-bed reference level. The models were obtained by regression against suspended sediment data from eight

different rivers and six experimental studies. The data cover a wide range of bed material grain sizes (44- 517 μm) and flow depths (0.06-32 m) and include grain-size specific data with up to 60 classes. Our empirical analysis suggests that sediment concentration increases with the ratio between shear velocity and settling velocity ($u_{*skin}/w_s$ or $u_*/w_{si}$) and Froude number

Earth **Surface**
**Dynamics**
Discussions

– both parameters also emerge as the key controlling variables in a forward model based on bedload layer concentrations. A parameter such as $u_*/w_{si}$, which represents the ratio of fluid force to particle settling, was also present in previous relations,

and its relevance reflects the competing effects of turbulent entrainment and particle settling. The Froude number dependence is less intuitive; it could be due to stronger sediment-induced stratification in large low-Fr rivers, but our forward model suggests that it emerges because of Fr-controls on bedload layer concentrations. Our preferred Rouse parameter model for the shape of the concentration profile suggests that sediment concentration is better mixed in the water column with larger $u_{*skin}/w_{si}$ and larger bed friction coefficient ($C_f$). The Rouse number is not inversely proportional to $u_*/w_{si}$, unlike standard

Rousean theory, indicating that sediment is more stratified than expected with $u_*/w_{si} > \sim 7$ and more well mixed than expected with $u_*/w_{si} < \sim 7$, possibly due to the competing effects of sediment-induced stratification when absolute concentrations are large and enhanced turbulent mixing when concentration gradients are steep. The dependence of Rouse number on bed friction coefficient might result from increased turbulence close to the bed in rivers with large bed roughness or bedforms. We also demonstrated that near-bed concentrations can be accurately predicted with saltation equations that have

been tested previously for gravel, suggesting a unified framework to model sand and gravel transport in rivers.

**Competing interests**

The authors declare that they have no conflict of interest.

**Data availability**

All data used in the analysis are provided in Table S1 of the Supplementary Materials. The MATLAB codes for the data analysis are available upon request.

**Author Contributions**

MPL and GP conceived the study. JdL compiled data and led data analysis with input from MPL. JdL and MPL wrote the initial manuscript. AJM, DH, JGV and JAN supplied suspended sediment data and contributed to the final manuscript.

**Acknowledgements**

We thank everyone who has been involved in the suspended sediment data collection in the Yellow River and the Fraser River (including Brandee Carlson, Hongbo Ma, Nicola Rammell, Kate Donkers, Jacqui Brown, Michael Wong, Michelle Linde, and Alex Gitto). This research was sponsored by the National Science Foundation (grant  ) to MPL, JAN and GP. A.J.M. was supported by a NSF Graduate Research Fellowship under Grant No. 1842494.



**Supplementary materials**

Table S1 – Contains all the suspended sediment data that was used to find the empirical relation

Table S2 – All entrainment and Rouse number models ranked according to goodness of fit as indicated by $r^2$.

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

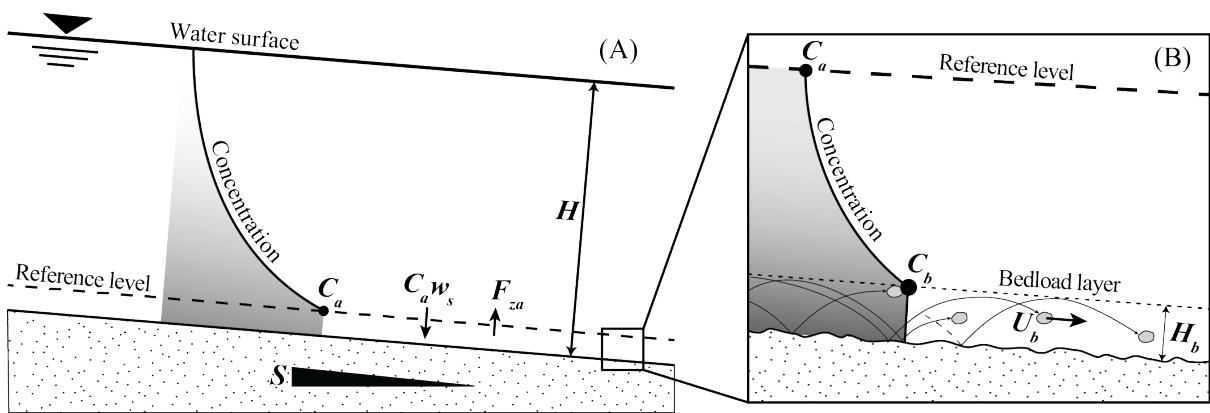

**Figure 1: (a) Definition diagram of suspended profile. (b) Extrapolation from saltation layer to reference level.**





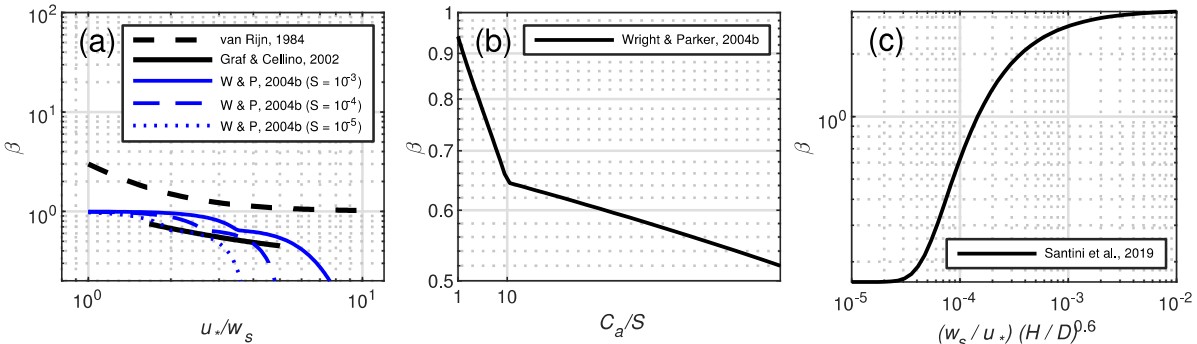

**Figure 2: Existing relations for $\beta$.** The relations predict that $\beta$ is a function of (a) $u_*/w_s$, (b) $c_a/S$ or (c) $(w_s/u_*$ (h / D)$^{0.6}$. In panel (a), to assess the dependence of the Wright & Parker (2004) model on $u_*/w_s$, $C_a$ is predicted with the entrainment relation from Wright & Parker (2004).

| (a) | Entrainment relations | | | |
|---|---|---|---|---|
| Author | Formula | Parameters | Reference height | Grain-size distribution |
| Einstein, 1950 | $E_s = \dfrac{1}{23.2}\dfrac{q_*}{(\tau_{*skin})^{0.5}}$ <br><br> $q_* = 3.97(\tau - 0.0495)^{1.5}$ | $\tau_{*skin}, D$ | $2D$ | Uniform |
| Engelund & Fredsøe, 1976 | $E_s = \dfrac{0.65}{(1+\lambda_b^{-1})^3}$ <br><br> $\lambda_b = \left[\dfrac{\tau_{*skin} - 0.06 - \dfrac{\beta p\pi}{6}}{0.027(R+1)\tau_{*skin}}\right]^{0.5}$ <br><br> $p = \left[1 + \left(\dfrac{\dfrac{\pi}{6}}{\tau_{*skin} - 0.06}\right)^4\right]^{-0.25}$ <br><br> $\beta = 1$ | $\tau_{*skin}$ | $2D$ | Uniform |
| Smith & McLean, 1977 | $E_s = \dfrac{0.65\gamma_0 S_0}{1+\gamma_0 S_0}$ <br><br> $with: S_0 = \dfrac{\tau_{*skin} - \tau_{*c}}{\tau_{*c}}$ <br><br> $and: \gamma_0 = 2.4 \cdot 10^{-3}$ <br><br> The Critical shields number ($\tau_{*c}$) is calculated with the Brownlie (1981) fit to the Shields curve: <br><br> $\tau_{*c} = 0.22 Re_p^{-0.6} + 0.06 \cdot 10^{(-7.7Re_p^{-0.6})}$ | $\tau_{*skin}; \tau_{*c}$ | $\alpha_0(\tau_{skin}^* - \tau_c^*)D + k_s$ <br> $\alpha_0 = 26.3$ | Uniform |
| Van Rijn, 1984 | $E_s = 0.015\dfrac{D}{a}\dfrac{S_0^{1.5}}{D_*^{0.3}}$ <br><br> $with: S_0 = \dfrac{\tau_{*skin} - \tau_{*c}}{\tau_{*c}}$ | $D, S, k_s$ | If known: 0.5x bedform height. | Uniform |





| | | | Else: $k_s$ with a minimum of $0.01H$ | |
|---|---|---|---|---|
| | $and: D_* = D \left(\frac{gR}{v}\right)^{1/3}$ | | | |
| Celik & Rodi, 1984 | $E_s = \frac{k_0 \bar{C}}{I}$ $\bar{C} = 0.034\left[1-\left(\frac{k_s}{H}\right)^{0.06}\right]\frac{u_*^2}{gRH}\frac{U}{w_s}$ $I = \int_{0.05}^{1}\left(\frac{1-\eta}{\eta}\cdot\frac{\eta_b}{1-\eta_b}\right)^{w_s/0.4u_*}d\eta$ $k_0 = 1.13, \eta = \frac{z}{H}, \eta_b = 0.05$ | $D, k_s, u_*, U, w_s$ | $0.05H$ | Uniform |
| Akiyama & Fukushima, 1986 | $E_s = 0; Z < Z_c$ $E_s = 3\times10^{-12}Z^{10}\left(1-\frac{Z_c}{Z}\right); Z_c < Z < Z_m$ $Z = \frac{u_*}{w_s}Re_p^{0.5}$ $Z_c = 5; \; Z_m = 13.2$ | $u_*, w_s, Re_p$ | $0.05H$ | Uniform |
| Garcia & Parker, 1991 | $E_{si} = \frac{A(\lambda X_i)^5}{1+\frac{A}{0.3}(\lambda X_i)^5}$ $with: X_i = \left(\frac{u_{*skin}}{w_{si}}Re_{pi}^{0.6}\right)\left(\frac{D_i}{D_{50}}\right)^{0.2}$ $A = 1.3\cdot10^{-7}$ $\lambda = 1 - 0.288\sigma_\phi$ $\sigma_\phi$ is the standard deviation of the bed sediment on the sedimentological $\phi$ scale | $u_{*skin}, w_{si}, Re_{pi}, D_i/D$ | $0.05H$ | Mixtures |
| McLean, 1992 | $E_{si} = E_s p_{sbi}$ $With:$ $E_s = \frac{0.65\gamma_0 S_0}{1+\gamma_0 S_0},$ $S_0 = \frac{\tau_{*skin}-\tau_{*c}}{\tau_{*c}}$ $\gamma_0 = 0.004$ $p_{sbi} = \frac{\varphi_i f_i}{\sum_{n=1}^{N}\varphi_i f_i}$ $\varphi_i = 1 \; for: u_* > w_{si}$ $\varphi_i = \frac{u_*-u_{*cr}}{w_{si}-u_{*cr}} \; for: u_* < w_{si}$ $u_{*c} = \sqrt{\frac{\tau_{*c}}{\rho}}$ | $\tau_{*c}, \tau_{*skin}, u_*, w_{si}$ | See original publication | Mixtures |
| Wright & Parker, 2004 | $E_{si} = \frac{B(\lambda X_i)^5}{1+\frac{B}{0.3}(\lambda X_i)^5}$ $with: X_i = \left(\frac{u_{*skin}}{w_{si}}Re_{pi}^{0.6}\right)S^{0.08}\left(\frac{D_i}{D_{50}}\right)^{0.2}$ | $u_{*skin}, w_{si}, Re_{pi}, S, D_i/D_{50}$ | $0.05H$ | mixtures |





| | $B = 7.8 \cdot 10^{-7}$ | | | |
|---|---|---|---|---|
| | $\lambda = 1 - 0.28\sigma_\phi$ | | | |
| | $\sigma_\phi$ is the standard deviation of the bed sediment on the sedimentological $\phi$ scale | | | |


| **(b)** | **Relations for β** | |
|---|---|---|
| ***Source*** | ***Equation*** | ***Parameters*** |
| van Rijn, 1984 | $\beta = 1 + 2\left(\dfrac{w_s}{u_*}\right)^2 \ \ for\ 0.1 < \dfrac{w_s}{u_*} < 1$ | $u_*/w_s$ |
| Graf & Cellino, 2002 | $\beta = \dfrac{3}{10} + \dfrac{3}{4}\dfrac{w_s}{u_*} \ \ for\ 0.2 < \dfrac{w_s}{u_*} < 0.6\ and\ no\ bedforms$ <br><br> $\beta = 1 + 2\left(\dfrac{w_s}{u_*}\right)^2 \ \ for\ 0.1 < \dfrac{w_s}{u_*} < 1\ and\ bedforms$ | $u_*/w_s$ |
| Wright & Parker, 2004 | $\beta = \begin{cases} 1 - 0.06\left(\dfrac{C_a}{S}\right)^{0.77} & for\ \dfrac{C_a}{S} \leq 10 \\ 0.67 - 0.0025\left(\dfrac{C_a}{S}\right) & for\ \dfrac{C_a}{S} > 10 \end{cases}$ | $C_a/S$ |
| Santini et al., 2019 (modified from Rose & Thorne, 2001) | $\beta = 3.1\ exp\left[-0.19 \times 10^{-3}\dfrac{u_*}{w_s}\left(\dfrac{H}{D}\right)^{0.6}\right] + 0.16$ | $u_*/w_s$, $h/D$ |

**Table 1: Existing formulas for Es (a) and β (b). The Critical shields number ($\tau_{*c}$) is calculated with the Brownlie (1981) fit to the Shields curve: $\tau_{*c} = 0.22Re_p^{-0.6} + 0.06 \cdot 10^{\left(-7.7Re_p^{-0.6}\right)}$**





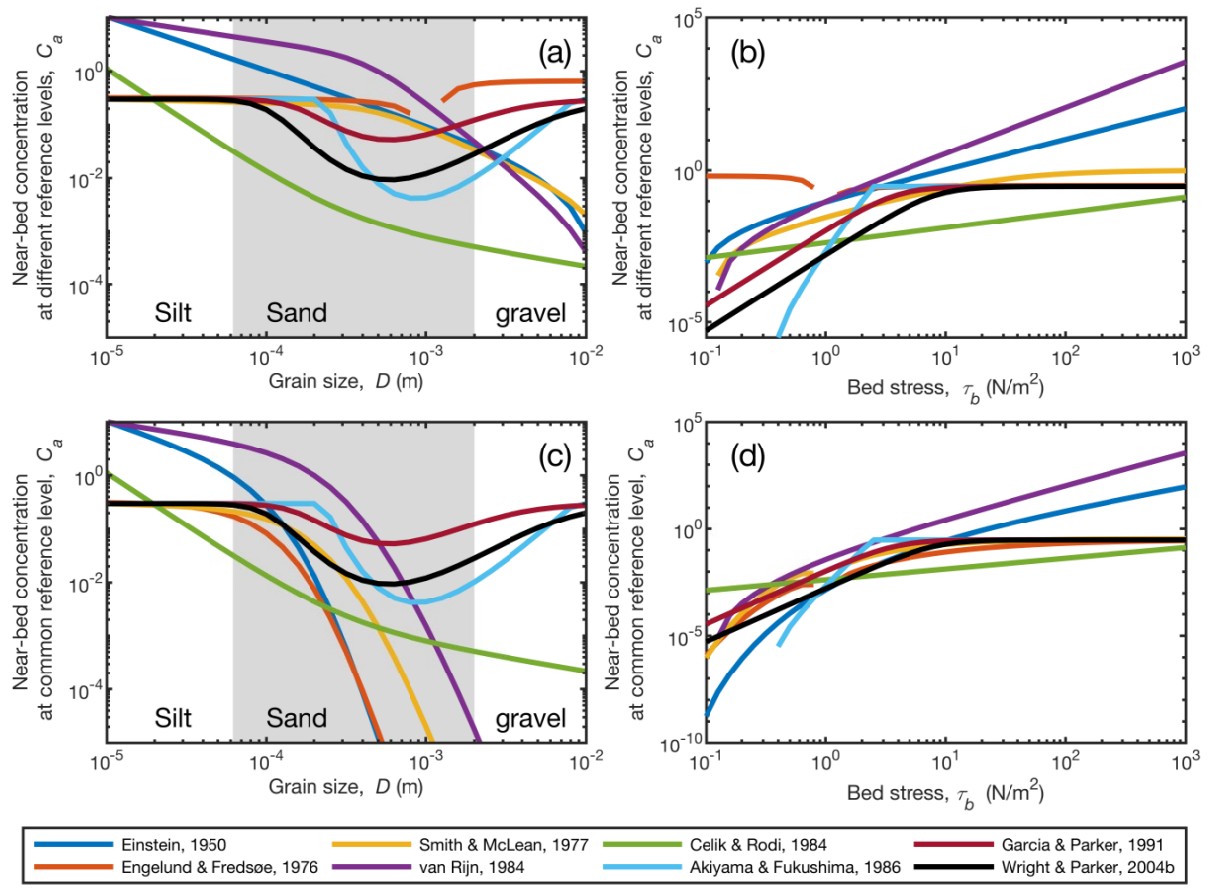

**Figure 3: Dependence of predicted near-bed concentration on grain size (a) and bed stress (b) for different previous entrainment relations. In panels (c) and (d) the predicted concentration is interpolated to a common reference level at 5% of the flow depth.**

| Data source | Type | Location | Median bed material grain size, $D_{50}$ (μm) | Water depth, H (m) | Number of grain size classes | Number of profiles |
|---|---|---|---|---|---|---|
| Jordan [1965] | river | Mississippi at St Louis | 189-457 | 3.54-16.34 | 12 | 51 |
| Nitrouer et al. [2011] | river | Mississippi at Empire reach | 166-244 | 12.96-32.38 | 43 | 9 |
| Lupker et al. [2011] | river | Ganges at Harding bridge | 159-268 | 10.0-14.0 | 31 | 7 |
| Nordin & Dempster [1963] | river | Rio Grande | 166-439 | 0.2-0.78 | 12 | 23 |





| | | | | | | |
|---|---|---|---|---|---|---|
| (unpublished) | river | Yellow River | 44-112 | 1.55-7.65 | 51 | 35 |
| (unpublished) | river | Fraser River | 300 | 8.7-14.5 | 60 | 25 |
| Hubbell & Matejka [1959] | river | Middle Loup River | 313-517 | 0.33-1.19 | 10 | 20 |
| Colby & Hembree [1955] | river | Niobrara River | 226-305 | 0.24-0.7 | 7 | 10 |
| Brooks [1954] | experiments | - | 160 | 0.059-0.085 | 1 | 7 |
| Barton & Lin [1955] | experiments | - | 180 | 0.091-0.42 | 1 | 29 |
| Coleman [1981] | experiments | - | 105-400 | 0.17 | 1 | 3 |
| Lyn [1986] | experiments | - | 150-240 | 0.065 | 1 | 3 |
| Sumer [1996] | experiments | - | 130 | 0.1 | 1 | 3 |
| Cellino [1998] | experiments | - | 135-210 | 0.12 | 1 | 17 |

**Table 2: Summary of experimental and field datasets included in the database.**

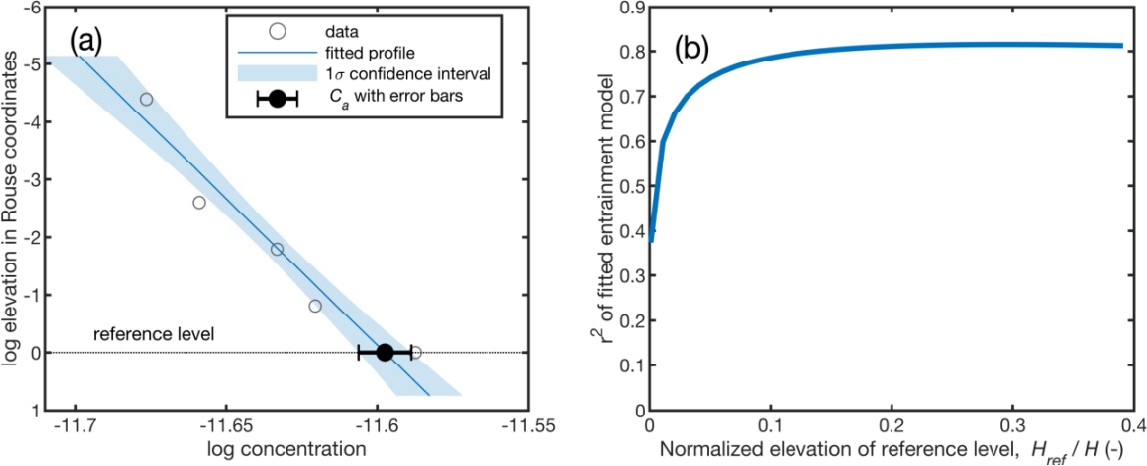

**Figure 4: (a) An example of suspended sediment data with a fitted Rouse profile. Extrapolation to the reference level at a gives the reference concentration ($C_a$) (b) Effect of reference level ($H_{ref}$) elevation on r² of entrainment relation.**


| Number of model parameters | Ranking in category | r² | Parameter 1 (P1) | Parameter 2 (P2) | Parameter 3 (P3) | Constant (A) | Exponent 1 (e1) | Exponent 2 (e2) | Exponent 3 (e3) |
|---|---|---|---|---|---|---|---|---|---|
| | 1 | 0.325 | $u_*/w_{si}$ | - | - | 9.91E-01 | -4.53E-01 | - | - |





| | | | P1 | P2 | P3 | A | e1 | e2 | e3 |
|---|---|---|---|---|---|---|---|---|---|
| **1** | 2 | 0.225 | $u_{*skin}/w_{si}$ | - | - | 7.18E-01 | -3.72E-01 | - | - |
| **2** | 1 | 0.396 | $u_{*skin}/w_{si}$ | $C_f$ | - | 1.45E-01 | -4.59E-01 | -3.00E-01 | - |
| | 2 | 0.378 | $u_*/w_{si}$ | $C_f$ | - | 3.87E-01 | -4.38E-01 | -1.61E-01 | - |
| **3** | 1 | 0.426 | $u_{*skin}/w_{si}$ | $C_f$ | $C_a/S$ | 2.85E-01 | -5.14E-01 | -2.12E-01 | 8.11E-02 |
| | 2 | 0.39 | $u_*/w_{si}$ | $C_f$ | $H/D$ | 2.74E-01 | -4.59E-01 | -1.41E-01 | 4.85E-02 |

Table 3: Best fitting one, two and three parameter models for Rouse number. The model relations have the following form: $P_i = A$ $P1^{e1} P2^{e2} P3^{e3}$. The shaded rows are the best-fit models for each number of parameters.

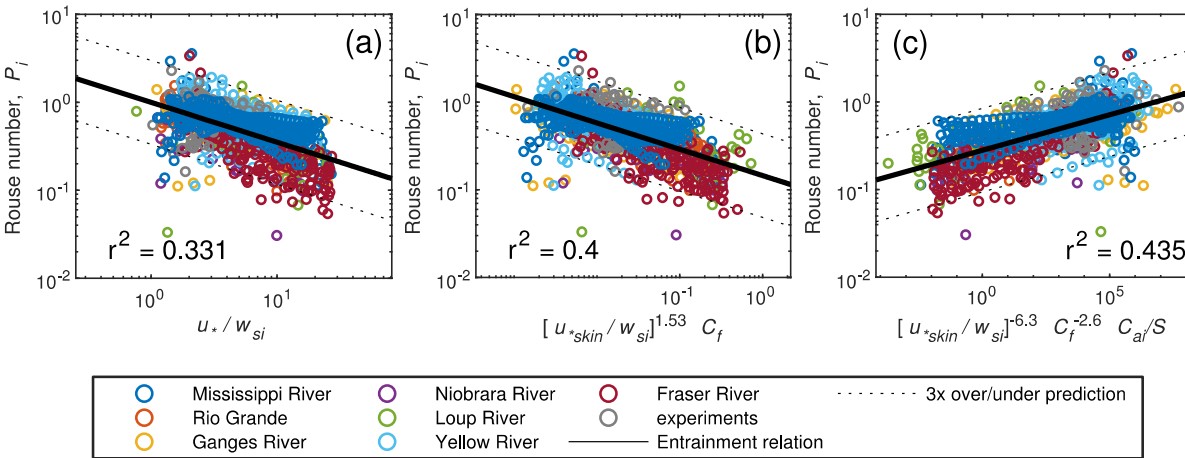

Figure 5: Best-fit models for Rouse number for grain-size specific data with one parameter (a), two parameters (b) and three parameters (c). The best-fit models correspond to the equations in Table 3 that are highlighted in grey. The equations of the best-fit models are rewritten such that the exponent on the last parameter becomes 1. For example, Eq. (23) on panel b is written as $P_i = \left( 0.145^{1/-0.3} \left( \frac{u_{*skin}}{w_{si}} \right)^{-0.46/-0.3} C_f \right)^{-0.3}$. This allows us to plot the data in the same way as Garcia & Parker (1991) and Wright & Parker (2004b).





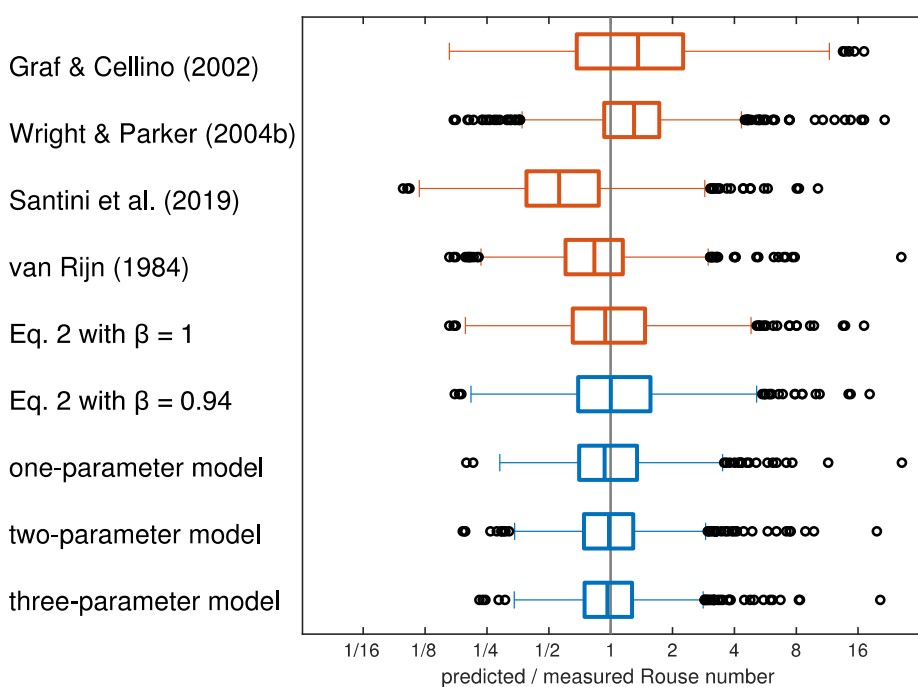

**Figure 6: Boxplot of measured versus predicted Rouse number for our models and previous models. The best-fit one, two and three parameter models correspond to the equations in Table 4 that are highlighted in grey.**

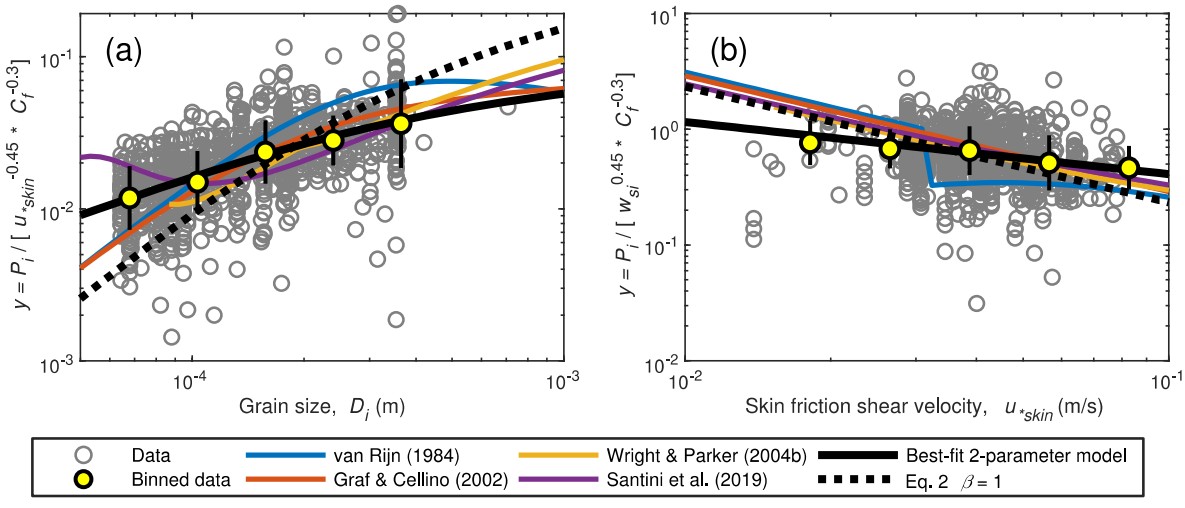

**Figure 7: Dependence of normalized Rouse number on grain size (a) and skin-friction shear velocity (b). Yellow filled symbols are the geometric mean and the error bars indicated the geometric standard deviation within each bin.**






| Number of model parameters | Ranking in category | $r^2$ | Parameter 1 (P1) | Parameter 2 (P2) | Parameter 3 (P3) | Constant (A) | Exponent 1 (e1) | Exponent 2 (e2) | Exponent 3 (e3) |
|---|---|---|---|---|---|---|---|---|---|
| **1** | 1 | 0.61 | $u_{*skin}/w_{si}$ | - | - | 4.23E-05 | 1.94E+00 | - | - |
| | 2 | 0.33 | $u_*/w_{si}$ | - | - | 3.66E-05 | 1.44E+00 | - | - |
| **2** | 1 | 0.79 | $u_{*skin}/w_{si}$ | $Fr$ | - | 4.74E-04 | 1.77E+00 | 1.18E+00 | - |
| | 2 | 0.74 | $u_*/w_{si}$ | $Fr$ | - | 7.04E-04 | 1.71E+00 | 1.81E+00 | - |
| **3** | 1 | 0.80 | $u_*/w_{si}$ | $Fr$ | $R_{pi}$ | 5.732E-03 | 1.31E+00 | 1.59E+00 | -8.60E-01 |
| | 2 | 0.79 | $u_{*skin}/w_{si}$ | $Fr$ | $S$ | 2.934E-04 | 1.75E+00 | 1.31E+00 | -7.93E-02 |

**Table 4: Best fitting models for entrainment ($E_{si}$) with one, two and three parameters. The fitted relations take the following form:** $E_{si} = A \, P1^{e1} \, P2^{e2} \, P3^{e3}$

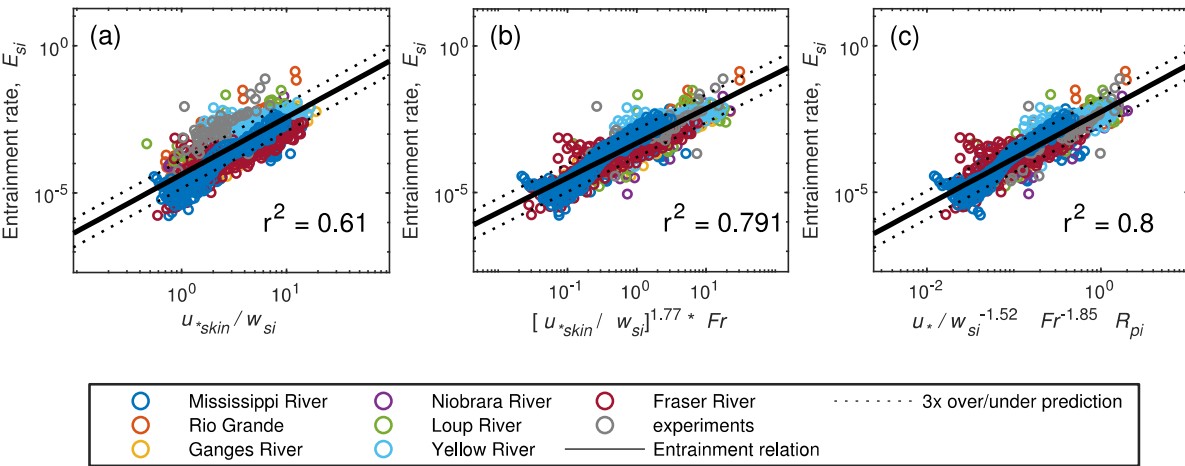

**Figure 8: Best fitting models for entrainment with one parameter (a), two parameters (b) and three parameters (c). The best-fit models correspond to the equations that are highlighted in grey in Table 4. The equations of the best-fit models are rewritten such**
**that the exponent on the last parameter becomes 1. This allows us to plot the data in the same way as Garcia & Parker (1991) and Wright & Parker (2004b).**





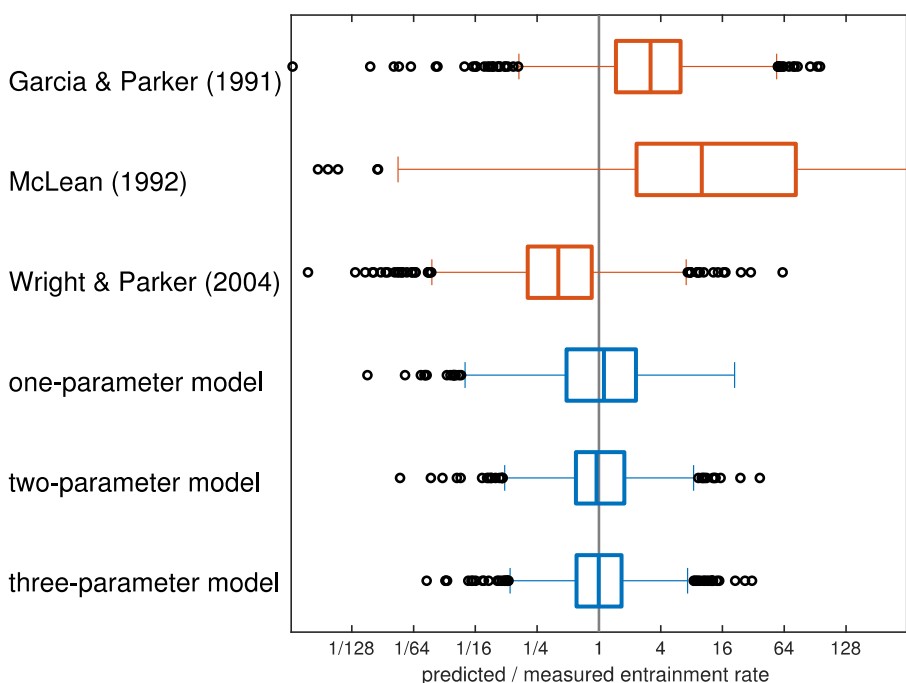

**Figure 9: Boxplot of predicted-to-measured ratio for our entrainment relations, $E_{si}$, and other relations. The best-fit one, two and three parameter models in this plot correspond to the equations that are highlighted in grey in Table 4.**

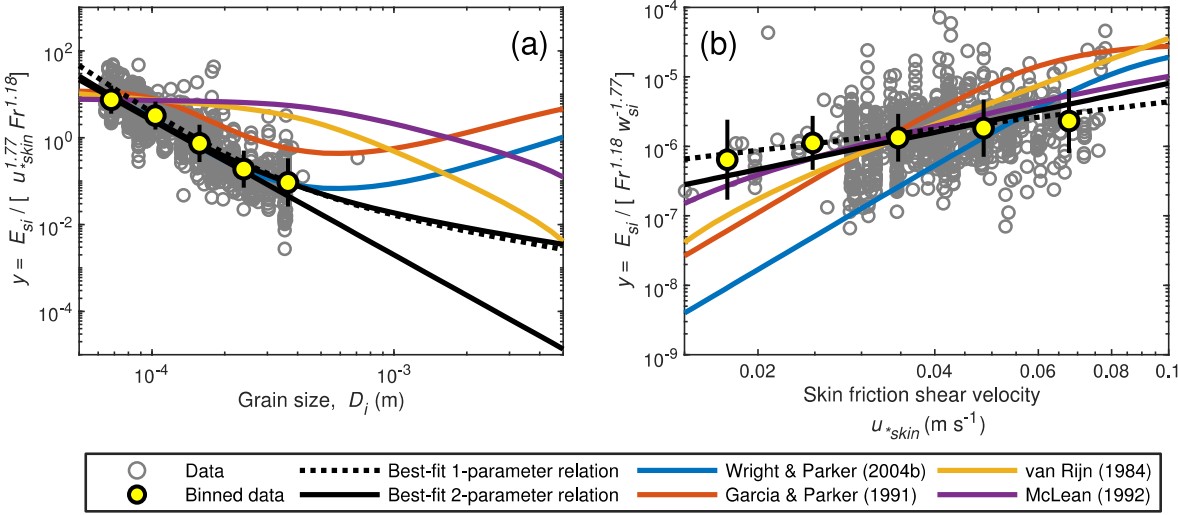

**Figure 10: Dependence of entrainment on grain size (a), and bed stress (b). Yellow filled symbols are the geometric mean and error bars indicate the geometrical standard deviation within each bin.**

| $r^2$ | Entrainment model | | | | | |
|---|---|---|---|---|---|---|
| | Best fit 1-parameter | Best fit 1-parameter | Best fit 2-parameter | Best fit 2-parameter | Best fit 3-parameter | Best fit 3-parameter |






| | | model containing $u_*/w_{si}$ | model containing $u_{*skin}/w_{si}$ | model containing $u_*/w_{si}$ | model containing $u_{*skin}/w_{si}$ | model containing $u_*/w_{si}$ | model containing $u_{*skin}/w_{si}$ |
|---|---|---|---|---|---|---|---|
| **Rouse model** | Best fit 1-parameter model containing $u_*/w_{si}$ | 0.646 | 0.768 | 0.862 | 0.873 | 0.878 | 0.871 |
| | Best fit 1-parameter model containing $u_{*skin}/w_{si}$ | 0.676 | 0.786 | 0.859 | 0.867 | 0.872 | 0.865 |
| | Best fit 2-parameter model containing $u_*/w_{si}$ | 0.634 | 0.768 | 0.856 | 0.876 | 0.878 | 0.876 |
| | Best fit 2-parameter model containing $u_{*skin}/w_{si}$ | 0.659 | 0.785 | 0.853 | 0.875 | 0.875 | 0.875 |
| | Best fit 3-parameter model containing $u_*/w_{si}$ | 0.637 | 0.757 | 0.802 | 0.818 | 0.819 | 0.817 |
| | Best fit 3-parameter model containing $u_{*skin}/w_{si}$ | 0.637 | 0.766 | 0.849 | 0.868 | 0.870 | 0.868 |

**Table 5: Goodness of predictions of grain-size specific sediment concentration at each sample elevation in the water column with different combinations of entrainment and Rouse models. The coefficient of determination ($r^2$) for each model combination is calculated for the log transformed data. Our proposed model for concentration in the water column combines the two-parameter entrainment model and the one-parameter Rouse model. This model combination is highlighted in grey in the table.**

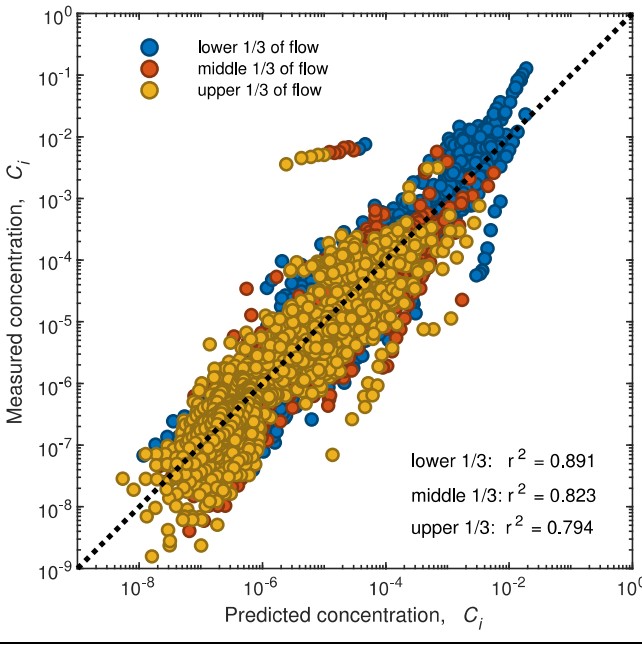





**Figure 11: Measured versus predicted grain-size specific concentration for each sample in the water column. Colors of the points indicate the relative elevation in the water column of each sample.**

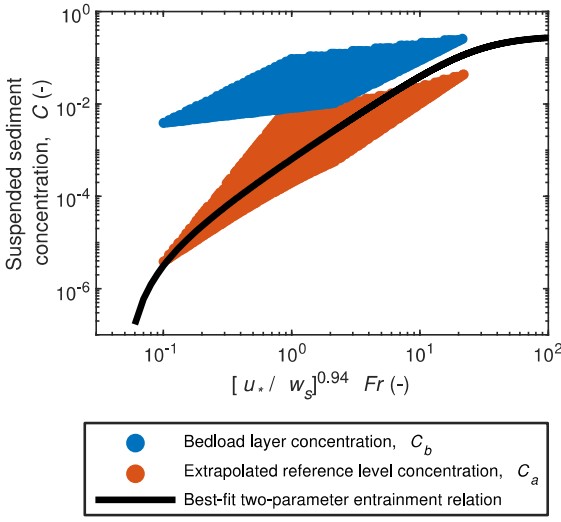

**Figure 12. Our proposed entrainment relation with overlain synthetic gravel data that represent results for the full range of parameter space: $0.1 < Fr < 1$ and $1 < \tau_* < 1000$.**

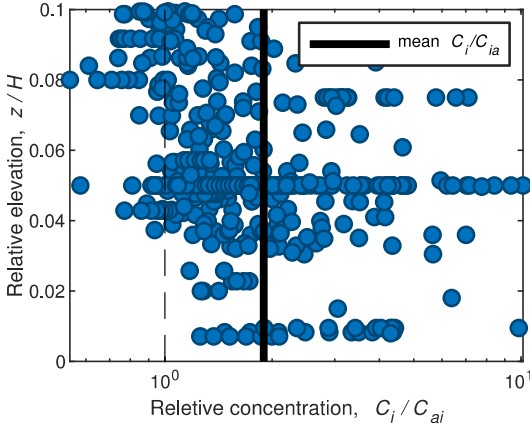


**Figure 13: Grain-size specific concentration data from the lowest 10% of the flow (i.e. below the reference level) plotted as a function of relative elevation. The average ratio of concentration in this domain is $C_i/C_{ai} = 1.9$.**





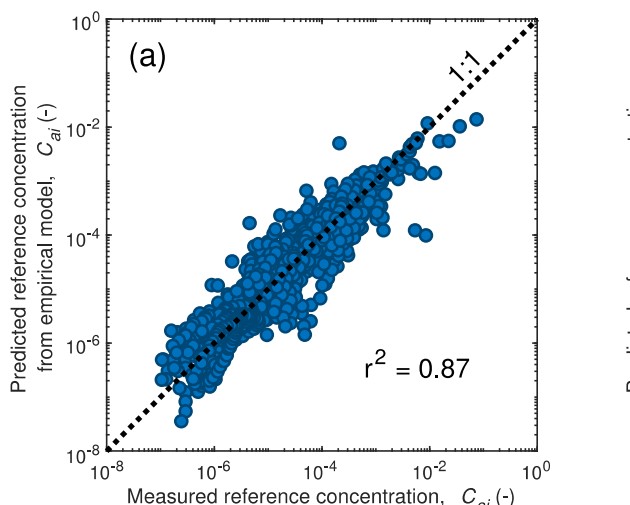
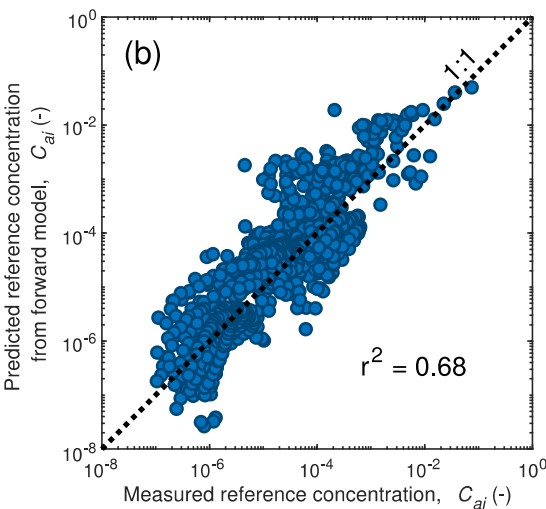

**Figure 14: Measured versus predicted grain-size specific entrainment rate for our empirical relation (a) and for the forward modelling approach (b).**
