# Peer review of "Entrainment and suspension of sand and gravel"

_Earth Surface Dynamics, 2019_

## Referee Comment (RC1) · Anonymous Referee #1 · 18 Jan 2020

Summary. The authors present a study on suspended sediment concentration via the Rouse equation. They review existing equations and through a new compilation of suspended sediment profiles they provide an improved empirical fit for the sediment entrainment function, a fitting parameter Beta, and the Rouse number.

General Comments. Overall the paper appears technically sound and I have no major reservations with the work presented by the authors that would present it from being published following some revisions. I have some general editorial or stylistic suggestions for the authors that may improve the manuscript and I leave it to them to implement them or not. As the introduction currently reads the paper appears focused on providing a better empirical fit to data than past empirically based equations. Rather than focusing on previous empirical equations, why not focus on the data and allow

your analysis to drive the narative. As an example, seeing all of the lines in Figure 3 is not that useful as some of them likely only differ due to differences in the datasets they were calibrated on. As is, I did not find the introduction to be any more insightful than that of Garcia and Parker (1991) other then adding a few more equations. It might be worthwhile to replace figures 2 & 3 with the concentration profiles and show the newly compiled data that is what really sets the current work apart from previous iterations.

The title in this regard seems a bit misleading as this paper is primarily about sand. The gravel component is interesting, however it is not as well integrated into the manuscript and may be better as a stand alone manuscript once data is available to validate the claims. I am not necessarily suggesting that it be removed, just that from my perspective it isn't the best fit at the moment given the data limitations and scope of the rest of the paper.

Specific Comments.

Ln. 97 - Could you provide the rational for beta=1. Lines 58-76 are all about beta being less than or greater than 1, but not equal. It is fine that it is one, but please work that reasoning into the preceeding paragraphs on beta.

Table 1a. The parameter column could use a bit more explanation or consistency. As an example, Smith & McLean 1977 have t*skin & t*c in the parameter column while Van Rijn, 1984 does not, even though they are both listed in the equation.

Figure 3. You might consider making this figure viewable in black and white or for people with visual impairments (color blindness).

Ln. 118 - 'workers' is a bit of an odd word for researchers here.

Ln. 227 - missing an 'is' or substract 'that'. '...one that based on...'

Ln. 252 - A shields stress of a 1000 seems to be a bit far fetched for gravel. Consider that at a 10% slope for pea gravel (∼0.5 cm) that would require an ∼80 m deep flow. That isn't realistic.

[Figure]

Ln. 258 - It is not clear that an R2 of 0.4 is significantly better than 0.33. The distributions of the predicted/measured (fig. 6) also do not look to be statistically distinct to make a claim of significance either. Could you instead provide some physical reasoning as to why the two parameter model is the best choice.

Fig.4 - Please clarify if the following is the correct interpretation. Equation (2) is fit to the profile data where P is treated as a fitting parameter. Then P is regressed against a variety of variables in Figure 5. This could be made a bit clearer in the begining of the results section as it was not entirely clear where P comes from in Fig. 5.

Fig. 7 - Could you provide a reasoning for the choice of binned data width and number of bins?

Fig. 5, 7, 8, 10, 11, 14 - Consider plotting the data as a 2D density plot as this won't obscure the majority of the data. At the moment it is hard to see what the data actually look like when they are all plotted on top of each other.

Ln. 288 - The previous relations (and the new ones) are all semi empirical based on limited field data, it is not surprising that by increasing the data (especially the ranges) that new model fit to these data performs better overall. I am not sure the numerous model comparisons are really a necessary component for this paper.

Fig. 10 - Not clear what the solid black line that tracks the dashed black line is in panel (a).

Ln. 312 - Not seeing a fig. 11b.

Ln. 381 - It would be worth taking a look at the recently published work by Ashley et al. (2020) in Water Resources Research on 'Estimating bedload from suspended load..'.

Ln. 385 - It looks like Ci/Ca increases as z/H approaches 0. You might show that the trend is not significant and that would justify the mean, which looks a bit skewed high, potentially by some outliers. Maybe the median would be a better parameter.

**ESurfD**

---

## Referee Comment (RC2) · Anonymous Referee #2 · 18 Feb 2020

This paper presents the development of new equations for transport by suspension. The author fit the Rouse number and Entrainment parameter with a large data set, and ultimately they derive a general equation for concentration. The results seem to be very promising; nonetheless the model was calibrated but not validated. The paper is well written but I think it could be improved for clarity, especially in introduction and discussion. I propose minor revision; the authors will have no difficulty in answering the various point presented below.

Comments I found the introduction a bit confused. Instead of presenting general considerations on suspension (why it is important, what do we know, what are the limitations, what are the differences between lowland and mountain rivers...), you go straight in a presentation of limitations of existing mechanistic approaches through a very exhaustive literature review (congratulation for the review) and new analysis. In addition the title is a bit confusing because when mentioning "sand and gravel" we expect more consideration for suspension of coarse sand and gravel, and this aspect is not really developed (in the introduction but also in the paper where the data sets comprises fine sands only) which, in my opinion, reduces the scope of the paper to situations where suspension can freely develop from fine bed sediments. Finally, it takes time to really understand the objectives of the paper. For clarity it might have been more efficient to really explain the context and objectives in introduction and describe the equations limitations in a next part called for instance "review of the existing theory".. ? This is a suggestion, I let the authors decide how to arrange the paper, but the must improve the message in introduction.

Line 80: If Fz is the upward flux of sediment it is not clear how Fz/ws is dimensionless. Could you give the dimension each time you introduce a parameters?

Line 177: Eq.7 is not usual; maybe you can give a reference or explain how it was obtained?

Line 188: it is not straightforward: write the Shields stress with Eq11

Line 199: does hiding effects make sense for sands?

Lines 213-214: this sentence is not really clear but is essential for understanding the methodology. I understood that you fit P with the data and compare to variables? I suggest that you develop a bit more this methodological point to insist on the absence of spurious correlation in Figure 5.

Line 230: Because of the absence of data, the approach for gravels is purely conceptual. One can for instance question on the validity of Eq20 and 21 at high shear stress (was this aspect considered in the original paper)?

Line 278: it could be clearer to start this paragraph with : "Figure 8 plots..." and explain again the parameters tested. For instance what was the reference level used for Esi in

Figure 8?

Line 295-297: The way it is presented seems a bit arbitrary. Could you give a reference for that?

Lines 398-399: I suppose that the threshold is 0.015 in Eq. 26? Is there a figure where we can visualize this threshold effect?

Line 301: Equation 26 is made complex to limit its maximum to 0.33. In my opinion it's too bad to lose the equation aesthetics: you could keep a simplest form and just mention Esi<=0.3

Line 310: Why don't you give the definitive model (used for Fig 11)?

Line 316: This very short paragraph looks like more a discussion point (a perspective) than a real result.

Discussion: It could be worth discussing the model limitations (if any). For instance I have in mind the complex interactions that may exist with coarse gravel and cobles beds in Mountain Rivers.

Your model has been calibrated but not validated. A lot of data are available in the literature could they be used for validation? If not could you discuss what should (could) be done for a validation in future research.

Figure 14: This result is surprisingly good. How was measured Cai for the runs considered? And what do you obtained when comparing qb_meas and qb_cal?

---

## Author Comment (AC1) · 11 Apr 2020

We thank the two referees for their thorough and constructive comments. We agree with nearly all of their points and have revised the manuscript incorporating their suggestions. We were pleased to see that both referees' found the study to be technical sound, with mostly stylistic suggestions, and both suggested minor revisions. The most significant change is to the introduction, as both referees' requested more motivation and road mapping for the paper. Below we give detailed responses to the referees' comments.

Anonymous Referee #1

Summary. The authors present a study on suspended sediment concentration via the

[Figure]

Rouse equation. They review existing equations and through a new compilation of suspended sediment profiles they provide an improved empirical fit for the sediment entrainment function, a fitting parameter Beta, and the Rouse number.

General Comments. Overall the paper appears technically sound and I have no major reservations with the work presented by the authors that would present it from being published following some revisions. I have some general editorial or stylistic suggestions for the authors that may improve the manuscript and I leave it to them to imple ment them or not. As the introduction currently reads the paper appears focused on providing a better empirical fit to data than past empirically based equations. Rather than focusing on previous empirical equations, why not focus on the data and allow your analysis to drive the narative. As an example, seeing all of the lines in Figure 3 is not that useful as some of them likely only differ due to differences in the datasets they were calibrated on. As is, I did not find the introduction to be any more insightful than that of Garcia and Parker (1991) other then adding a few more equations. It might be worthwhile to replace figures 2 & 3 with the concentration profiles and show the newly compiled data that is what really sets the current work apart from previous iterations.

REPLY: We thank the reviewer for their thorough and constructive comments. We agree on almost all points. We have made several edits to the introduction following the reviewer's suggestion. We now better motivate the study and discuss why entrainment is important, what we know and what are the limitations. And we also better roadmap the paper. It's not clear to us how to use the data to motivate the work, because we do not present the data until after going through the methods. One of the main motivations for the work is the inconsistency in previous work. Because of this, we decided to leave the different models on Figure 3. Reviewer # 2 seems to agree that the review of previous models is useful.

The title in this regard seems a bit misleading as this paper is primarily about sand. The gravel component is interesting, however it is not as well integrated into the manuscript and may be better as a stand alone manuscript once data is available to validate the

claims. I am not necessarily suggesting that it be removed, just that from my perspective it isn't the best fit at the moment given the data limitations and scope of the rest of the paper.

REPLY: This is a good point. We understand that data on gravel is limited; however, we do present theory for gravel and compare the gravel theory to the sand data. This is a main point of our paper—to develop a theory for entrainment that may be applicable to both sand and gravel. The need for such a theory has been emphasized in the revised introduction. We present theory for gravel saltation in the Methods (Section 2.3), results (section 3.4) and discussion (Section 4.3). For these reasons we prefer to leave gravel in the title. We hope our paper will help show a path forward in future work to test the relation under the gravel regime.

Specific Comments. Ln. 97 - Could you provide the rational for beta=1. Lines 58-76 are all about beta being less than or greater than 1, but not equal. It is fine that it is one, but please work that reasoning into the preceeding paragraphs on beta.

REPLY: This has been fixed. Betta = 1 is a common assumption, and is the simplest model.

Table 1a. The parameter column could use a bit more explanation or consistency. As an example, Smith & McLean 1977 have t*skin & t*c in the parameter column while Van Rijn, 1984 does not, even though they are both listed in the equation.

REPLY: This has been corrected.

Figure 3. You might consider making this figure viewable in black and white or for people with visual impairments (color blindness).

REPLY: We have changed the line thickness to further distinguish the models. Thanks for this suggestion. We also carried this through to Figure 10.

Ln. 118 - 'workers' is a bit of an odd word for researchers here.

REPLY: Changes as suggested.

Ln. 227 - missing an 'is' or substract 'that'. '...one that based on...'

REPLY: Changes as suggested.

Ln. 252 - A shields stress of a 1000 seems to be a bit far fetched for gravel. Consider that at a 10% slope for pea gravel ( 0.5 cm) that would require an 80 m deep flow. That isn't realistic.

REPLY: Thank you for this point. This is simply a range over which to explore the model. Fine sand and silt in mountain rivers can have shields numbers this high. They can also occur during geologically significant floods, such as the in the Missoula Floods. We list references in the main text.

Ln. 258 - It is not clear that an R2 of 0.4 is significantly better than 0.33. The distributions of the predicted/measured (fig. 6) also do not look to be statistically distinct to make a claim of significance either. Could you instead provide some physical reasoning as to why the two parameter model is the best choice.

REPLY: We have clarified in the text that we prefer the two parameter model because the r2 is not significantly improved with the addition of a third parameter, and that there are many three parameter models with similar misfits. That said, all models are given in Table S2 including their r2 values, so the reader can choose which ever model best fits the application.

Fig.4 - Please clarify if the following is the correct interpretation. Equation (2) is fit to the profile data where P is treated as a fitting parameter. Then P is regressed against a variety of variables in Figure 5. This could be made a bit clearer in the begining of the results section as it was not entirely clear where P comes from in Fig. 5.

REPLY: Yes, that is correct. We have added further clarification on line 270 - 275.

Fig. 7 - Could you provide a reasoning for the choice of binned data width and number
of bins?

REPLY: This is an arbitrary choice for binned width. This has been added to the figure captions.

Fig. 5, 7, 8, 10, 11, 14 - Consider plotting the data as a 2D density plot as this won't obscure the majority of the data. At the moment it is hard to see what the data actually look like when they are all plotted on top of each other.

REPLY: We appreciate this point. We tried plotting the data many different ways, but decided the best way is to bin data to show the central tendency. We have added binned data to Figures 5, 8 and 14. We believe that Fig. 11 would be confusing with binned data since it is already divided into three groups. Figs. 7 and 10 already had binned data.

Ln. 288 - The previous relations (and the new ones) are all semi empirical based on limited field data, it is not surprising that by increasing the data (especially the ranges) that new model fit to these data performs better overall. I am not sure the numerous model comparisons are really a necessary component for this paper.

REPLY: We understand this point and thank you for the opportunity to clarify. Although the models are semi-empirical, they are intended to represent the physics in a generic way. One of the motivations for us to conduct this study is that the previous papers did not show comparisons between different models. This makes it unclear how the various entrainment models compare with each other. This is why we think it is important to show how the model compares to previous models.

Fig. 10 - Not clear what the solid black line that tracks the dashed black line is in panel (a).

REPLY: This was an error and has been deleted. Thank you for noting that.

Ln. 312 - Not seeing a fig. 11b.

[Figure]

REPLY: This typo has been fixed.

Ln. 381 - It would be worth taking a look at the recently published work by Ashley et al. (2020) in Water Resources Research on 'Estimating bedload from suspended load...'.

REPLY: Thank you for this suggestion. We have added that citation as suggested.

Ln. 385 - It looks like Ci/Ca increases as z/H approaches 0. You might show that the trend is not significant and that would justify the mean, which looks a bit skewed high, potentially by some outliers. Maybe the median would be a better parameter.

REPLY: We have added binned data that more clearly shows that the concentration data is nearly uniform with depth. We agree and have added the geometric mean value rather than the mean.

Anonymous Referee #2

This paper presents the development of new equations for transport by suspension. The author fit the Rouse number and Entrainment parameter with a large data set, and ultimately they derive a general equation for concentration. The results seem to be very promising; nonetheless the model was calibrated but not validated. The paper is well written but I think it could be improved for clarity, especially in introduction and discussion. I propose minor revision; the authors will have no difficulty in answering the various point presented below. Comments I found the introduction a bit confused. Instead of presenting general considerations on suspension (why it is important, what do we know, what are the limitations, what are the differences between lowland and mountain rivers. . .), you go straight in a presentation of limitations of existing mechanistic approaches through a very exhaustive literature review (congratulation for the review) and new analysis. In addition the title is a bit confusing because when mentioning "sand and gravel" we expect more consideration for suspension of coarse sand and gravel, and this aspect is not really developed (in the introduction but also in the

paper where the data sets comprises fine sands only) which, in my opinion, reduces the scope of the paper to situations where suspension can freely develop from fine bed sediments. Finally, it takes time to really understand the objectives of the paper. For clarity it might have been more efficient to really explain the context and objectives in introduction and describe the equations limitations in a next part called for instance "review of the existing theory".. ? This is a suggestion, I let the authors decide how to arrange the paper, but the must improve the message in introduction.

REPLY: Thank you for raising this concern about the introduction. We have made several edits to the introduction following the reviewer's suggestion. We now better motivate the study and discuss why entrainment is important, what we know and what are the limitations. And we also better roadmap the paper as suggested.

Line 80: If Fz is the upward flux of sediment it is not clear how Fz/ws is dimensionless. Could you give the dimension each time you introduce a parameters?

REPLY: We have added dimensions to the parameters as suggested.

Line 177: Eq.7 is not usual; maybe you can give a reference or explain how it was obtained?

REPLY: We have rewritten this equation in its typical form, added a reference, and better explained how it is used. Thank you for pointing out the confusion.

Line 188: it is not straightforward: write the Shields stress with Eq11

REPLY: Thank you for pointing this out. We have added the algebra to make this substitution.

Line 199: does hiding effects make sense for sands?

REPLY: This parameter was included by Garcia and Parker, 1991; Wright and Parker, 2004. We are not aware of an argument against hiding effects in sands.

Lines 213-214: this sentence is not really clear but is essential for understanding the

methodology. I understood that you fit P with the data and compare to variables? I suggest that you develop a bit more this methodological point to insist on the absence of spurious correlation in Figure 5.

REPLY: We expanded this sentence to clarify the point.

Line 230: Because of the absence of data, the approach for gravels is purely conceptual. One can for instance question on the validity of Eq20 and 21 at high shear stress (was this aspect considered in the original paper)?

REPLY: We appreciate this concern. It is partly theoretical, but partly based on semi-empirical equations from datasets on gravel saltation. At very high shear stresses, Eq. (20) and (21) asymptote to reasonable values (Chatanantavet et al., 2013). We added these points to revised paper.

Line 278: it could be clearer to start this paragraph with : "Figure 8 plots. . ." and explain again the parameters tested. For instance what was the reference level used for Esi in Figure 8?

REPLY: This has been changed as suggested.

Line 295-297: The way it is presented seems a bit arbitrary. Could you give a reference for that?

REPLY: Yes, this is following the approach of Garcia and Parker (1991).

Lines 398-399: I suppose that the threshold is 0.015 in Eq. 26? Is there a figure where we can visualize this threshold effect?

REPLY: We believe the reviewer is referring to lines 298-299 here. Yes, we clarified in the text that the threshold is 0.015 and added the model with the threshold to Fig. 8b.

Line 301: Equation 26 is made complex to limit its maximum to 0.33. In my opinion it's too bad to lose the equation aesthetics: you could keep a simplest form and just mention Esi<=0.3

REPLY: We appreciate this point, but have followed previous work on framing this limiting value.

Line 310: Why don't you give the definitive model (used for Fig 11)?

REPLY: Good point. We have added the model to the text here and in the caption to Fig. 11.

Line 316: This very short paragraph looks like more a discussion point (a perspective) than a real result.

REPLY: We understand this point. However, because we manipulated the bedload transport equations, as discussed in the Methods, we think it is appropriate to show the results of this work in the Results. We have added text for the gravel part of the study to the introduction and methods to better integrate this work into the paper.

Discussion: It could be worth discussing the model limitations (if any). For instance I have in mind the complex interactions that may exist with coarse gravel and cobles beds in Mountain Rivers.

REPLY: We have added unknown limitations with complex topography in mountain rivers on line 498

Your model has been calibrated but not validated. A lot of data are available in the literature could they be used for validation? If not could you discuss what should (could) be done for a validation in future research.

REPLY: We have added the need for model validation on line 460.

Figure 14: This result is surprisingly good. How was measured Cai for the runs considered? And what do you obtained when comparing qb_meas and qb_cal?

REPLY: Cai is concentration at 10% of the flow depth from our large dataset used throughout the results. This has been clarified in the figure caption. Unfortunately we cannot compare the total sediment flux because we did not develop a sediment or flow

velocity model.
* * *